# AssembleFlow: Rigid Flow Matching with Inertial Frames for Molecular Assembly

**Hongyu Guo**
National Research Council Canada
University of Ottawa
`hongyu.guo@uottawa.ca`

**Yoshua Bengio**
Mila - Québec AI Institute
Université de Montréal
CIFAR AI Chair
`yoshua.bengio@mila.ca`

**Shengchao Liu**
Université de Montréal
`shengchao.liu@umontreal.ca`

## Abstract

Molecular assembly, where a cluster of rigid molecules aggregated into strongly correlated forms, is fundamental to determining the properties of materials. However, traditional numerical methods for simulating this process are computationally expensive, and existing generative models on material generation overlook the rigidity inherent in molecular structures, leading to unwanted distortions and invalid internal structures in molecules. To address this, we introduce AssembleFlow. AssembleFlow leverages inertial frames to establish reference coordinate systems at the molecular level for tracking the orientation and motion of molecules within the cluster. It further decomposes molecular SE(3) transformations into translations in $\mathbb{R}^3$ and rotations in SO(3), enabling explicit enforcement of both translational and rotational rigidity during each generation step within the flow matching framework. This decomposition also empowers distinct probability paths for each transformation group, effectively allowing for the separate learning of their velocity functions: the former, moving in Euclidean space, uses linear interpolation (LERP), while the latter, evolving in spherical space, employs spherical linear interpolation (SLERP) with a closed-form solution. Empirical validation on the benchmarking data COD-Cluster17 shows that AssembleFlow significantly outperforms six competitive deep learning baselines by at least 45% in assembly matching scores while maintaining 100% molecular integrity. Also, it matches the assembly performance of a widely used domain-specific simulation tool while reducing computational cost by 25-fold.

## 1 Introduction

Deep learning methods have been revolutionizing scientific research across various domains, enabling breakthroughs in fields such as drug discovery (Yu et al., 2024), material science (Merchant et al., 2023), and molecular design (Loeffler et al., 2024). For instance, protein folding systems have demonstrated unprecedented accuracy and creativity in designing protein structures (Jumper et al., 2021; Baek et al., 2021), driving innovation in drug discovery. These advancements underscore the transformative potential of machine learning in tackling complex scientific problems.

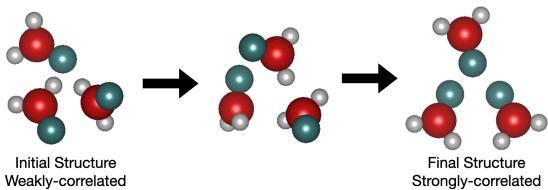

Initial Structure
Weakly-correlated

Final Structure
Strongly-correlated

Figure 1: Illustration of the assembly of a cluster of three molecules transitioning from a weakly correlated structure (left) to a strongly correlated crystal structure (right). A key challenge for existing generative models in material generation is preserving the **rigidity** of each molecule throughout this transformation in 3D space.

Molecular assemble or crystallization is one such complex process where rigid molecules transition from a weakly correlated arrangement to a highly ordered, strongly correlated structure. During this

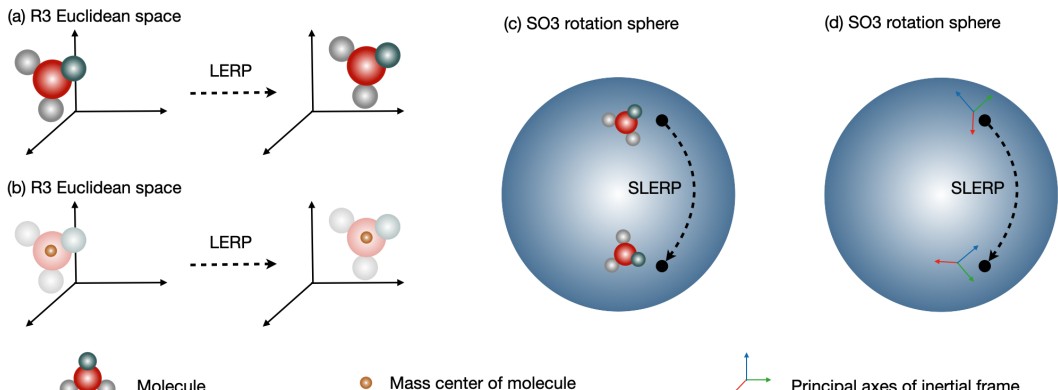

Figure 2: (a, b): The molecule's center of mass (CoM) is used as the reference point for **translation** and interpolation in Euclidean space. (c, d): An inertial frame serves as the reference coordinate system, tracking rigid molecular **rotation**—where rotation operations are implemented using quaternion representation—and enabling interpolation in spherical space.

process, each molecule is approximated to maintain its shape and structure unchanged as it moves in the 3D space, as illustrated in Figure 1. Crystallization plays a pivotal role in determining the physical properties of materials, including their mechanical strength, electrical conductivity, and thermal stability (Porter et al., 2009; Carter & Norton, 2013), making it a key process in material science (Ashby & Jones, 2012), pharmaceuticals (Hilfiker, 2006), and nanotechnology (Gonsalves et al., 2000). For example, the crystalline form of a drug can affect its solubility and bioavailability (Byrn et al., 1999; Healy et al., 2017); Similarly, precise control over molecular arrangements is key to optimizing the electronic and catalytic performance of organic semiconductors, polymers, and molecular catalysts (Saparov & Mitzi, 2016).

Traditional numerical methods have long been employed to simulate the crystallization process (Martínez et al., 2009; Van Der Spoel et al., 2005), but they are often computationally expensive and inefficient, limiting their scalability and practical use in large-scale applications. On the other hand, despite the importance of crystallization, existing machine learning methods struggle to capture the physical constraints critical to this process. A major limitation is the failure to account for the inherent rigidity of molecular structures during crystallization (Liu et al., 2024c), often leading to unwanted distortions and invalid internal structures, *i.e.*, non-rigid molecules. In an assembly, molecules must retain their rigid atomic structures, as this rigidity is essential for producing meaningful packing arrangements. However, current generative models for molecular crystallization treat molecules as flexible entities (Liu et al., 2024c), resulting in physically unrealistic packing structures and atomic arrangements, failing to retain individual molecule's structure intact.

To address these limitations, we introduce AssembleFlow, a novel framework specifically designed to incorporate the rigid body constraints inherent in molecular assembly or crystallization. As illustrated in Figure 2, AssembleFlow leverages inertial frames to establish reference coordinate systems for assembling molecules. Because SE(3) is the semi-direct product of the rotation group SO(3) and the translation group $\mathbb{R}^3$, we can further decompose the group SE(3) transformations into translations in $\mathbb{R}^3$ (Figure 2 (a, b)) and rotations in SO(3) (Figure 2 (c, d)). Such decomposition allows the explicit enforcement of both translational and rotational rigidity at the molecular level effectively, ensuring that each molecule in the cluster moves as a unified, rigid body throughout the crystallization process. During such enforcement, AssembleFlow employs a distinct approach for learning the SE(3)-equivariant velocity functions associated with translations and rotations. For translations, it uses linear interpolation (LERP) in Euclidean space, while for rotations, it leverages spherical linear interpolation (SLERP) in spherical space, with a closed-form solution. This distinction in handling the translation and rotation groups allows AssembleFlow to accurately model the rigid transformations of molecules during each prediction and generation step within the flow matching framework (Lipman et al., 2022; Liu et al., 2022; Albergo & Vanden-Eijnden, 2022).

We empirically evaluate AssembleFlow using the benchmarking crystallization dataset COD-Cluster17. The quantitative results reveal that AssembleFlow significantly outperforms six competitive deep learning baselines by at least 45% in terms of assembly matching score. Also, AssembleFlow

exhibits strong assembly performance compared to a widely used domain-specific simulation tool for molecular assembly, achieving this with a 25-fold reduction in computational cost. Furthermore, we present qualitative results, including atomic collision properties of predicted crystals, which further demonstrate AssembleFlow's effectiveness in preserving and modeling the rigidity of the molecular crystallization and assembly process. Our work is the first to implement rigid generation in SE(3) space for molecular assembly. We also want to mention that in what follows, we use molecular assembly, crystallization, and molecular packing interchangeably.

## 2 PRELIMINARIES

**Molecular crystallization.** Molecular crystallization is a transition of molecules from weakly correlated structures to strongly correlated structures, *e.g.*, from liquid or gas phase to solid phase, as illustrated in Figure 1. One example is liquid water freezing into ice, transitioning from a liquid phase to a solid phase. The crystallization from a gas phase directly to a solid phase is called deposition.

**SE(3)-equivariance.** For geometric modeling for crystallization, one critical property of the target function is rotation-equivariant and translation-equivariant (*i.e.*, SE(3)-equivariant). We here provide a brief introduction on the SE(3)-equivariance, and for more detailed discussions of SE(3)-equivariance, we refer the reader to (Smidt et al., 2018; Brandstetter et al., 2021; Liu et al., 2023; Zhang et al., 2023). SE(3)-equivariance is the property for the geometric modeling function $f : X \to Y$ as:

$$f(\rho_X(\boldsymbol{a})\boldsymbol{x}) = \rho_Y(\boldsymbol{a})f(\boldsymbol{x}), \ \ \forall \boldsymbol{a} \in G, \boldsymbol{x} \in X, \tag{1}$$

where $\rho_X(\boldsymbol{a})$ and $\rho_Y(\boldsymbol{a})$ are the SE(3) group representations on the input and output space, respectively. SE(3)-equivariant modeling in Equation (1) is essentially saying that the designed deep learning model $f$ is modeling the whole SE(3) group transformation trajectory on the molecule conformations, and the output is the transformed $\hat{y}$ accordingly. One concrete example is that when we rotate the input molecular system by a certain angle, the predicted forces by SE(3)-equivariant models will also rotate accordingly.

**Conditional flow matching.** Conditional flow matching (CFM) (Lipman et al., 2022) and two parallel works (Rectified Flow (Liu et al., 2022) and Stochastic Interpolants (Albergo & Vanden-Eijnden, 2022)) formulate the distribution modeling problem as learning a vector field that can generate a probability path mapping from simple distribution at $t = 0$ to the target distribution at $t = 1$. Please refer to the original papers for a more detailed discussion (Lipman et al., 2022).

In the crystallization processes, our geometric data are atomic coordinates in the 3D Euclidean points $\boldsymbol{r} \in \mathbb{R}^3$, and the **atomic type is fixed** during the whole crystallization process, so we may as well ignore that. Then we define time-dependent vector field $\boldsymbol{v} : [0, 1] \times \mathbb{R}^3 \to \mathbb{R}^3$. A time-dependent vector field defines a time-dependent diffeomorphic map, called flow, $\phi : [0, 1] \times \mathbb{R}^3 \to \mathbb{R}^3$. The vector field defines flow via an ordinary differential equation as

$$\frac{d\phi_t(\boldsymbol{r})}{dt} = v_t(\phi_t(\boldsymbol{r})). \tag{2}$$

A probability density path is denoted as $p : [0, 1] \times \mathbb{R}^3 \to \mathbb{R}_{>0}$. Existing flow model (Chen et al., 2018) maps a prior distribution $p_0$ to another distribution $p_t$ with push-forward equation or change of variable rule: $p_t(\boldsymbol{r}) = [\phi_t]_* p_0(\boldsymbol{r}) = p_0(\phi_t^{-1}(\boldsymbol{r})) \det \left| \frac{d\phi_t(\boldsymbol{r})}{dx} \right|^{-1}$. Thus, modeling the likelihood of data distribution at $t = 1$ can be transformed into modeling the velocity field matching problem with parameterized velocity field $v_\theta$, *i.e.*, flow matching:

$$\mathcal{L}_{\text{FM}} = \mathbb{E}_{t,\boldsymbol{r}} \big\| v_t(\boldsymbol{r}) - v_\theta(\boldsymbol{r}, t) \big\|^2. \tag{3}$$

With the continuity equation (Villani et al., 2009), we can further derive an equivalent objective by considering the conditional vector field conditioned on the empirical data $\boldsymbol{r}_1$, *i.e.*, $v_t(\boldsymbol{r}|\boldsymbol{r}_1)$, and the resulting objective is the conditional flow matching:

$$\mathcal{L}_{\text{CFM}} = \mathbb{E}_{t,\boldsymbol{r},\boldsymbol{r}_1} \big\| v_t(\boldsymbol{r}|\boldsymbol{r}_1) - v_\theta(\boldsymbol{r}, \boldsymbol{r}_1, t) \big\|^2. \tag{4}$$

## 3 METHOD: ASSEMBLEFLOW

**Problem Formulation.** AssembleFlow is designed to model rigid transformations during crystallization, ensuring that each molecule in the cluster remains rigid throughout the transformation process.

Rigorously, we are modeling $P(\{\boldsymbol{r}_f\}|\{\boldsymbol{r}_i\})$, where $\{\boldsymbol{r}_f\}$ and $\{\boldsymbol{r}_i\}$ are the atom conformations in final and initial positions respectively. During this process, we assume the rigidity of molecules. Noticeably, we will use a preprocessed dataset where the prior conformations are geometrically optimized and fixed (Liu et al., 2024c).

This section outlines the five key steps in the algorithm's development. Specifically, in Section 3.1, we explain how inertial frames can be leveraged to provide a stable reference for tacking the orientation of multiple assembling molecules in the Euclidean space. Such a reference perspective can guarantee rigid structures during molecular rotations throughout the assembly process. Building on these frames, there are multiple ways for rotation representation, so in Section 3.2, we illustrate how to use quaternion representation for capturing the rotation transformation induced from inertial frames. This is followed by a detailed discussion of AssembleFlow, a rigid flow matching method, in Section 3.3. AssembleFlow decomposes the assembly probabilistic paths into SO(3) group path and $\mathbb{R}^3$ group path to guarantee the rigidity, and learns the time-dependent vector fields through a flow-matching framework on the two path spaces respectively. In Section 3.4, we employ the reparameterization trick to make AssembleFlow more numerically stable. Finally, in Section 3.5, we present the two types of SE(3)-equivariant flow matching velocity functions specifically designed for use in AssembleFlow. Note: the pseudo algorithm of our AssembleFlow is provided in Appendix E.3.

## 3.1 SO(3) GROUP AND INERTIAL FRAME FOR RIGID PACKING

The core of AssembleFlow lies in the utilization of the inertial frame as the reference frame. Within this frame, the rotation matrix in the SO(3) group defines how the molecular system rotates rigidly. This serves as the key step in AssembleFlow for modeling rigid transformations in SO(3) group.

**SO(3) group.** The special orthogonal group, denoted as SO($n$), is a group of rotation matrices that represent rotations in $n$-dimensional Euclidean space. In this paper, we are interested in $n = 3$ dimensional space, and every rotation matrix used to perform a rotation in 3D space can be represented as an element of SO(3) group. The SO(3) group consists of all orthogonal matrices with determinant 1 $R \in \mathbb{R}^{3 \times 3}$ such that $R^T R = I$, where $R^T$ is the transpose of $R$ and $I$ is the identity matrix.

**Inertial frame as the reference frame.** An inertial frame is a reference frame such that it can provide a consistent basis for describing a molecule's motion, including rotation. Here, we utilize the inertial frame to build a basis for explaining how each molecule rotates in the Euclidean space. One example is illustrated in Figure 2. Importantly, an inertial frame provides a coordinate system such that a molecule stays rigid and does not deform over the crystallization or modeling process; we here assume the system is not influenced by external forces. Next, we will detail how inertial frames are used to represent the rotation matrix for rigid molecules.

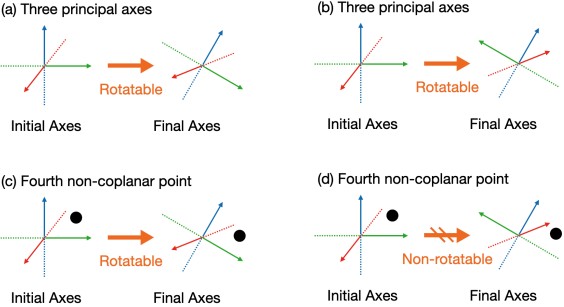

Figure 3: (a, b) show two potential rotational alignments between two coordinate systems (axes). (c, d) show that only one unique rotation is possible for four non-coplanar points.

First, we employ the following four sequential steps to derive the reference frames that construct the rotation matrix from $N$ atomic positions $\boldsymbol{r}$:

- Calculate the mass center: $\boldsymbol{c} = \frac{1}{N} \sum_i \boldsymbol{r}_i$.
- Adjust position relative to the center $\boldsymbol{r}_i = \boldsymbol{r}_i - \boldsymbol{c}$.
- Compute the inertia tensor $\hat{I} = \sum_i \|\boldsymbol{r}_i\|^2 I - \boldsymbol{r}_i \boldsymbol{r}_i^T$, where $I$ is the unit diagonal matrix.
- Obtain the principal axes of inertia by applying eigen-decomposition on $\hat{I}$. We have $\hat{I} = Q\Lambda Q^T$, where $Q$ is the orthogonal matrix whose columns are the eigenvectors of $\hat{I}$, and $\Lambda$ is the diagonal matrix whose elements are the eigenvalues $\lambda_i$ of $\hat{I}$, representing the principal moments of inertial along the principal axes.

The above steps yield the three principal axes in $Q$. To adopt this for modeling the crystallization process (Figure 1), we build inertial frames for each molecule in the cluster. In our case, for each

molecule we need to build two inertial frames, for the weakly correlated and strongly correlated structures, respectively. We call these two frames **initial (inertial) frame** $\mathcal{F}_i$ and **final (inertial) frame** $\mathcal{F}_f$.

Second, we apply the eigen-decomposition to obtain the **initial principal axes** $Q_i$ and **final principal axes** $Q_f$, respectively. As illustrated in Figure 2 (d), we can only perform the rotation based on the aligned principal axes or the aligned coordinate systems. However, as we conduct the eigen-decomposition of frames $\mathcal{F}_i$ and $\mathcal{F}_f$, it is not guaranteed that the corresponding principal axes $Q_i$ and $Q_f$ are aligned. To align the two coordinate systems, we aim to match both the directions and the orders of the corresponding principal axes in each set. We will detail this alignment process next.

**Align the directions between initial and final coordinate systems.** First, for a given inertial frame $\mathcal{F}$, we have three axes in $Q$ composing a coordinate system. This can lead to eight possible directions. To align the directions between the initial system $Q_i$ and final system $Q_f$, we add a first constraint – the three axes must form a right-handed coordinate system. Such a filtering step can be achieved by using cross-product: if the local frame system is right-handed, then the cross-product between any two axes should match the third axis or share the same direction as the third axis. Otherwise, the coordinate system is left-handed, then we randomly revert one basis out of three. This reduces to four potential combinations of directions. To further align the directions, we introduce Lemma 1 and Theorem 1 to help provide us with theoretical guidance.

**Lemma 1.** *For an initial inertial frame $\mathcal{F}_i$ and a final inertial frame $\mathcal{F}_f$, we build up the corresponding right-handed principal axes as coordinate systems, $Q_i$ and $Q_f$, respectively. Suppose we have to change the directions of $Q_f$ to match $Q_i$, then we should change the directions of two bases in $Q_f$.*

*Proof.* There are three bases in $Q_f$. If we change one or three basis directions in $Q_f$, then $Q_f$ will change from right-handedness to left-handedness, which violates the assumption. Thus, if we need to change the directions of $Q_f$ to match $Q_i$, we should change the directions of two bases in $Q_f$. □

With Lemma 1, we find that using three base vectors is insufficient to determine the direction alignment of three axes. To define the directions that can match between the initial and final coordinate systems, we need to incorporate an extra node as an auxiliary, as in Theorem 1.

**Theorem 1.** *For an initial inertial frame $\mathcal{F}_i$ and a final inertial frame $\mathcal{F}_f$, we build up the corresponding right-handed principal axes as coordinate systems, $Q_i$ and $Q_f$, respectively. Then we need to incorporate a fourth point that is not coplanar with the three basis vectors, to align the directions of two coordinate systems with one unique rotation transformation matrix.*

*Proof Sketch.* We first provide intuitive examples in Figure 3. In Figure 3 (a, b), we can see at least two possible rotation matrices to transform from initial axes to final axes. However, when we add a fourth non-coplanar point in Figure 3 (c), the rotation transformation becomes unique, and the corresponding rotation in Figure 3 (d) is invalid. Then more rigorously, the proof includes the two key steps: (1) Using Lemma 1, we can find multiple rotation matrices for alignment between coordinate systems. (2) After introducing the fourth non-coplanar point, the contradiction proves that there exists only one unique rotation for alignment. For more rigorous proof, please refer to Appendix D. □

**Align the ordering between initial and final coordinate systems.** We can typically sort the eigenvectors (as for principal axes) through the corresponding eigenvalues. The main challenge comes when there is a tie in eigenvalues. Because this is a rare case, we propose doing a depth-first-search to enumerate all the possible combinations of basis orderings of $Q_f$, to match $Q_i$.

**Outputs and engineering issue: numerical stability.** Without loss of generality, we can assume that we do not change the axis direction or ordering in the initial coordinate system $Q_i$, and we only change $Q_f$ to $\hat{Q}_f$, so as to align with $Q_i$. The ultimate rotation matrix is thus $R = Q_i^T \hat{Q}_f$. Meanwhile, we would like to point out that multiple numerical stability issues exist. This can arise in the following scenarios: (1) when the sampled points are near the origin, (2) when checking if the eigenvalues are tied or not, (3) when extracting a fourth non-planar point for alignment, (4) when verifying whether rotating the initial atoms (points) matches the final atoms. To mitigate these issues, we carefully select a threshold value and clamp the (reconstructed) coordinates to this minimum threshold.

**Summary.** In this section, we first introduce the basic concepts of $SO(3)$ group and inertial frame. Then we present how we construct the initial and final inertial frames for each molecule, *i.e.*, $\mathcal{F}_i$ and $\mathcal{F}_f$, in molecular crystallization. Next, by applying the eigen-decomposition on the constructed

inertial frames, we obtain the initial and final principal axes (right-handed), $Q_i$ and $Q_f$, respectively. Finally, we align the $\hat{Q}_f$ to $\hat{Q}_i$ by checking the directions and ordering of three axes in $Q_f$. This results in two aligned bases ($Q_i$ and $\hat{Q}_f$) and a rotation matrix $R$ such that $\hat{Q}_f = R Q_i$. As a result, this enables rigid molecule-level rotations during their transformations in the assembling processes.

## 3.2 QUATERNION REPRESENTATION FOR ROTATION

For the SO(3) group in Section 3.1, there are multiple ways to represent a rotation transformation in addition to the rotation matrix $R$. If we want to model the rotation matrix directly, we must guarantee that the generated matrix variable satisfies the two properties discussed in Section 3.1, namely Orthogonality and Determinant. Such a constrained modeling is challenging. Thus, an alternative way of rotation representation with a more flexible formulation is preferred. To attain this goal, we utilize quaternion representation defined through the inertial frame, as described below.

**Definition.** A quaternion $q$ is defined as:

$$q = w + xi + yj + zk = (w, v), \tag{5}$$

where $w, x, y, z$ are real numbers, and $i, j, k$ are the fundamental quaternion units. Equivalently, the $w$ is called the real part, while $v = (x, y, z)$ is a 3D vector representing the imaginary part.

**Rotation quaternion.** A rotation quaternion is a unitary quaternion, *i.e.*, $w^2 + x^2 + y^2 + z^2 = 1$, and this can be easily achieved by taking the normalization of the quaternion variables. In what follows, we will assume the quaternion is a rotation quaternion unless otherwise specified. Notice that for each rotation matrix, there are two equivalent quaternions, $q$ and $-q$. Here we manually enforce the real number part of the generated quaternion to be non-negative.

**Tranformation from rotation matrix to rotation quaternion.** There are multiple ways to extract the quaternion from the rotation matrix, and we provide a more detailed discussion in Appendix B. In this work, we adopt eigendecomposition (Horn, 1987; Bar-Itzhack, 2000) to extract the initial and final quaternion (*i.e.*, $q_i$ and $q_f$) from the initial and final coordinate systems (*i.e.*, $Q_i$ and $Q_f$).

**Spherical interpolation (SLERP) for quaternion interpolation.** One of the main advantages of using quaternion is that it is friendly to interpolation on the SO(3) space, *i.e.*, the spherical interpolation (SLERP) between two quaternions $q_0$ and $q_1$:

$$\text{SLERP}(q_0, q_1, t) = \frac{\sin((1 - t)\omega)q_0 + \sin(t\omega)q_1}{\sin(\omega)}, \tag{6}$$

where $\omega$ is the angle between $q_0$ and $q_1$, and $t \in [0, 1]$ is interpolation parameter. Thus, we can see that SLERP provides a smooth and uniform rotation between two quaternions. An example is provided in Figure 2.

We provide a comprehensive discussion of various rotation representations in Appendix B, including quaternion multiplication and vector rotation using quaternion. Please consult that section for details.

## 3.3 PATH INTERPOLATION IN ASSEMBLEFLOW

To model the crystallization process, our AssembleFlow method integrates the inertial frames and quaternion representation for rotation, as discussed in Sections 3.1 and 3.2, into a conditional flow matching framework. We note that, unlike most existing flow matching methods that focus solely on atom-level diffusion paths in the Euclidean space, which suffices for non-rigid transformation, AssembleFlow operates within the full SE(3) group space in the molecule level due to the rigidity requirement.

Recall that the crystallization process involves the movement over a cluster of molecules, and for each molecule in the cluster, AssembleFlow jointly models the rotational transformations in SO(3) space and translational transformations in $\mathbb{R}^3$ space. Such a decomposition ensures the preservation of rigid molecular structures throughout the crystallization process. We next detail these two transformations.

**Modeling translations in $\mathbb{R}^3$.** The goal is to model the molecule-level translations in each cluster, and AssembleFlow achieves this by modeling the translations on each molecule's mass center, as depicted in Figure 2. For notation simplicity, we will use $x \in \mathbb{R}^3$ to represent the translation vector.

We adopt the flow-matching framework, and the goal here is to learn the probability of final mass center $\boldsymbol{x}_f$ from the initial mass center $\boldsymbol{x}_i$, *i.e.*, $p(\boldsymbol{x}_f | \boldsymbol{x}_i)$. To this end, we assume that we use linear interpolation (LERP) for path interpolation, by treating $\boldsymbol{x}_0 = \boldsymbol{x}_i$ and $\boldsymbol{x}_1 = \boldsymbol{x}_f$, then for interpolation parameter $t \in [0, 1]$, we have the interpolated translation as:

$$\text{LERP}(\boldsymbol{x}_0, \boldsymbol{x}_1, t) = t\boldsymbol{x}_0 + (1 - t)\boldsymbol{x}_1. \tag{7}$$

Next, we introduce an SE(3)-equivariant function $\boldsymbol{v}_{\theta, \mathbb{R}^3}(\boldsymbol{x}_t, t)$ as the core module to learn the velocity at time $t$. Thus the objective function is defined as:

$$\mathcal{L}_{\mathbb{R}^3} = \|\boldsymbol{x}_1 - \boldsymbol{x}_0 - \boldsymbol{v}_{\theta, \mathbb{R}^3}(\boldsymbol{x}_t, \boldsymbol{q}_t, t)\|^2. \tag{8}$$

**Modeling rotations in SO(3).** For modeling the rotations in the SO(3) group, recall that we can find the initial bases $Q_i$ and final principal bases $Q_f$ from inertial frames in Section 3.1, and then we transform them into the rotation quaternions $\boldsymbol{q}_i$ and $\boldsymbol{q}_f$ as introduced in Section 3.2. The task here is to model $p(\boldsymbol{q}_f | \boldsymbol{q}_i)$.

Thus, it is natural to adopt the spherical interpolation (SLERP) as the smooth translation between two quaternions. We treat $\boldsymbol{q}_0 = \boldsymbol{q}_i$ and $\boldsymbol{q}_1 = \boldsymbol{q}_f$ and plug them into Equation (6). This gives us the interpolated rotations at time $t$. The first-order derivative of SLERP has an analytical formula:

$$\frac{d}{dt}\text{SLERP}(\boldsymbol{q}_0, \boldsymbol{q}_1, t) = \frac{\omega\Big(\cos(t\omega)\boldsymbol{q}_1 - \cos((1-t)\omega)\boldsymbol{q}_0\Big)}{\sin(\omega)}. \tag{9}$$

Similarly here, we then introduce an SE(3)-equivariant function $\boldsymbol{v}_{\theta, \text{SO}(3)}(\boldsymbol{q}_t, t)$ to model the molecule-level rotation velocity at time $t$. The objective function becomes:

$$\mathcal{L}_{\text{SO}(3)} = \|\frac{d}{dt}\text{SLERP}(\boldsymbol{q}_0, \boldsymbol{q}_1, t) - \boldsymbol{v}_{\theta, \text{SO}(3)}(\boldsymbol{x}_t, \boldsymbol{q}_t, t)\|^2. \tag{10}$$

**Inference.** For inference, AssembleFlow conducts the sampling step in SO(3) and $\mathbb{R}^3$ alternatively:

$$\boldsymbol{x}_{t+1} = \boldsymbol{x}_t + \delta t \cdot \boldsymbol{v}_{\theta, \mathbb{R}^3}(\boldsymbol{x}_t, \boldsymbol{q}_t, t), \qquad \boldsymbol{q}_{t+1} = \boldsymbol{q}_t + \delta t \cdot \boldsymbol{v}_{\theta, \text{SO}(3)}(\boldsymbol{x}_t, \boldsymbol{q}_t, t). \tag{11}$$

Thus, both the molecule-level translation $\boldsymbol{x}_{t+1}$ and molecule-level rotation $\boldsymbol{q}_{t+1}$ are applied on each molecule, and we repeat Equation (11) for $T$ steps to obtain the predicted strongly correlated molecule position. However, in Equation (11), it remains an open question on how to obtain the $\omega$ for SO(3) generation, since $\omega$ is the angle between $\boldsymbol{q}_0$ and $\boldsymbol{q}_1$, and $\boldsymbol{q}_1$ is unknown during the inference process. We address this by proposing the reparameterization trick as will be discussed next in Section 3.4.

## 3.4 Reparameterization for Strongly Correlated Structures

We here leverage the reparameterization trick to directly model the SE(3) action at time $T$ instead of the velocity at each time $t$. Equivalently, the velocity of SE(3) action can be written as:

$$\boldsymbol{v}_{\theta, \mathbb{R}^3}(\boldsymbol{x}_t, \boldsymbol{q}_t, t) = (\hat{\boldsymbol{x}}_{1, \theta}(\boldsymbol{x}_t, \boldsymbol{q}_t, t) - \boldsymbol{x}_t)/(1 - t),$$

$$\boldsymbol{v}_{\theta, \text{SO}(3)}(\boldsymbol{x}_t, \boldsymbol{q}_t, t) = \frac{\omega\Big(\cos(t\omega)\hat{\boldsymbol{q}}_{1, \theta}(\boldsymbol{x}_t, \boldsymbol{q}_t, t) - \cos((1-t)\omega)\boldsymbol{q}_0\Big)}{\sin(\omega)}. \tag{12}$$

In other words, we directly estimate the translation $\hat{\boldsymbol{x}}_{1, \theta}(\boldsymbol{x}_t, \boldsymbol{q}_t, t)$ and rotation $\hat{\boldsymbol{q}}_{1, \theta}(\boldsymbol{x}_t, \boldsymbol{q}_t, t)$ in the final step or the strongly correlated structure. The objectives over the two spaces thus become:

$$\mathcal{L}_{\mathbb{R}^3, \text{reparameter}} = \mathbb{E}[\|\boldsymbol{x}_1 - \hat{\boldsymbol{x}}_{1, \theta}(\boldsymbol{x}_t, \boldsymbol{q}_t, t)\|^2], \qquad \mathcal{L}_{\text{SO}(3), \text{reparameter}} = \mathbb{E}[\|\boldsymbol{q}_1 - \hat{\boldsymbol{q}}_{1, \theta}(\boldsymbol{x}_t, \boldsymbol{q}_t, t)\|^2]. \tag{13}$$

The final objective function is the summation of two terms. Besides, such a reparameterization enables us to conduct inference using the Euler algorithm:

$$\boldsymbol{x}_{t+1} = \boldsymbol{x}_t + \delta_t \cdot (\hat{\boldsymbol{x}}_{1, \theta}(\boldsymbol{x}_t, \boldsymbol{q}_t, t) - \boldsymbol{x}_t)/(1 - t),$$

$$\boldsymbol{q}_{t+1} = \boldsymbol{q}_t + \delta_t \cdot \frac{\hat{\omega}\Big(\cos(t\hat{\omega})\hat{\boldsymbol{q}}_{1, \theta}(\boldsymbol{x}_t, \boldsymbol{q}_t, t) - \cos((1-t)\hat{\omega})\boldsymbol{q}_0\Big)}{\sin(\hat{\omega})}, \tag{14}$$

where $\hat{\omega}$ is the angle between $\boldsymbol{q}_0$ and $\hat{\boldsymbol{q}}_{1, \theta}(\boldsymbol{x}_t, \boldsymbol{q}_t, t)$. The velocity functions of $\hat{\boldsymbol{x}}_{1, \theta}(\boldsymbol{x}_t, \boldsymbol{q}_t, t)$ and $\hat{\boldsymbol{q}}_{1, \theta}(\boldsymbol{x}_t, \boldsymbol{q}_t, t)$ are SE(3)-equivariant and will be discussed next in Section 3.5.

Table 1: AssembleFlow against six generative models on COD-Cluster17 with 5K, 10K, and all samples. The best results are marked in **bold**.

| | Packing Matching | | Validity | | |
|---|---|---|---|---|---|
| | PM (atom) ↓ | PM (center) ↓ | Collision ↓ | Separation ↑ | Compactness ↑ |
| Dataset: **COD-Cluster17-5K** | | | | | |
| GNN-MD | $13.67 \pm 0.06$ | $13.80 \pm 0.07$ | $27.53 \pm 0.49$ | $0.22 \pm 0.11$ | $\mathbf{100.00 \pm 0.00}$ |
| CrystalSDE-VE | $15.52 \pm 1.48$ | $16.46 \pm 0.99$ | $1.20 \pm 0.08$ | $27.17 \pm 0.86$ | $57.47 \pm 7.76$ |
| CrystalSDE-VP | $18.15 \pm 3.02$ | $19.15 \pm 4.46$ | $0.84 \pm 0.14$ | $53.13 \pm 12.89$ | $34.00 \pm 30.75$ |
| CrystalFlow-VE | $14.87 \pm 7.07$ | $13.08 \pm 4.51$ | $1.37 \pm 0.04$ | $35.70 \pm 0.73$ | $8.40 \pm 4.17$ |
| CrystalFlow-VP | $15.71 \pm 2.69$ | $17.10 \pm 1.89$ | $1.38 \pm 0.04$ | $35.43 \pm 0.88$ | $4.87 \pm 1.09$ |
| CrystalFlow-LERP | $13.59 \pm 0.09$ | $13.26 \pm 0.09$ | $0.34 \pm 0.01$ | $97.38 \pm 0.10$ | $\mathbf{100.00 \pm 0.00}$ |
| AssembleFlow (ours) | $\mathbf{7.27 \pm 0.04}$ | $\mathbf{6.13 \pm 0.10}$ | $\mathbf{0.33 \pm 0.00}$ | $\mathbf{97.64 \pm 0.36}$ | $\mathbf{100.00 \pm 0.00}$ |
| Dataset: **COD-Cluster17-10K** | | | | | |
| GNN-MD | $13.83 \pm 0.06$ | $13.90 \pm 0.05$ | $27.88 \pm 0.49$ | $0.23 \pm 0.11$ | $\mathbf{100.00 \pm 0.00}$ |
| CrystalSDE-VE | $17.25 \pm 2.46$ | $17.86 \pm 1.11$ | $0.99 \pm 0.27$ | $32.99 \pm 10.72$ | $34.93 \pm 14.99$ |
| CrystalSDE-VP | $22.20 \pm 3.29$ | $21.39 \pm 1.50$ | $0.53 \pm 0.35$ | $52.48 \pm 15.44$ | $16.83 \pm 18.09$ |
| CrystalFlow-VE | $16.41 \pm 2.64$ | $16.71 \pm 2.35$ | $1.42 \pm 0.03$ | $33.79 \pm 0.51$ | $5.47 \pm 0.47$ |
| CrystalFlow-VP | $19.39 \pm 4.37$ | $16.01 \pm 3.13$ | $1.44 \pm 0.03$ | $33.35 \pm 0.55$ | $4.23 \pm 0.48$ |
| CrystalFlow-LERP | $13.54 \pm 0.03$ | $13.20 \pm 0.03$ | $0.32 \pm 0.00$ | $97.32 \pm 0.05$ | $\mathbf{100.00 \pm 0.00}$ |
| AssembleFlow (ours) | $\mathbf{7.38 \pm 0.03}$ | $\mathbf{6.21 \pm 0.05}$ | $\mathbf{0.31 \pm 0.00}$ | $\mathbf{97.73 \pm 0.16}$ | $99.93 \pm 0.05$ |
| Dataset: **COD-Cluster17-All** | | | | | |
| GNN-MD | $22.30 \pm 12.04$ | $14.51 \pm 0.82$ | $24.29 \pm 4.58$ | $4.13 \pm 5.60$ | $98.77 \pm 1.73$ |
| CrystalSDE-VE | $17.28 \pm 0.73$ | $18.92 \pm 0.03$ | $0.19 \pm 0.18$ | $15.47 \pm 12.42$ | $2.51 \pm 2.37$ |
| CrystalSDE-VP | $18.03 \pm 4.56$ | $20.02 \pm 3.70$ | $0.55 \pm 0.19$ | $48.78 \pm 1.70$ | $6.88 \pm 2.82$ |
| CrystalFlow-VE | $12.80 \pm 1.20$ | $15.09 \pm 0.34$ | $1.41 \pm 0.01$ | $35.34 \pm 0.28$ | $2.90 \pm 0.02$ |
| CrystalFlow-VP | $13.50 \pm 0.44$ | $13.28 \pm 0.48$ | $1.51 \pm 0.02$ | $33.06 \pm 1.31$ | $6.61 \pm 3.17$ |
| CrystalFlow-LERP | $13.61 \pm 0.00$ | $13.28 \pm 0.01$ | $0.34 \pm 0.00$ | $97.34 \pm 0.02$ | $\mathbf{99.99 \pm 0.01}$ |
| AssembleFlow (ours) | $\mathbf{7.37 \pm 0.01}$ | $\mathbf{6.21 \pm 0.01}$ | $\mathbf{0.31 \pm 0.00}$ | $\mathbf{98.15 \pm 0.22}$ | $99.98 \pm 0.00$ |

## 3.5 SE(3)-EQUIVARIANT MULTI-GRAINED VELOCITY FUNCTION

Recall that the data structure considered here is the cluster of molecules, thus it is natural to split the modeling into intra-molecule and inter-molecule modeling, as introduced below. For **intra-molecule** modeling, we adopt the PaiNN (Schütt et al., 2021), which is one of the most widely used SE(3)-equivariant models. It can encode the inherent geometric structural information of individual molecules. Then for **inter-molecule** modeling, we consider two options of SE(3)-equivariant models: (1) Atomic-level modeling that utilizes all the atoms' positions for learning the molecular-level rotation and translation for the next step. (2) Molecular-level modeling that directly utilizes the molecular-level rotation and translation for next-step prediction. This concludes our discussion on AssembleFlow, and more details are provided in Appendix E. A high-level overview and pseudo algorithm are provided in Algorithms 1 and 2 in Appendix E.3.

## 4 EXPERIMENTS

## 4.1 EXPERIMENT SETUP

**Implementation.** The codes and checkpoints are available at this GitHub repository.

**Datasets.** We evaluate our method using the crystallization dataset COD-Cluster17 (Liu et al., 2024c). This COD-Cluster17 contains 133K crystals and is a curated subset derived from the Crystallography Open Database (COD) database (Grazulis et al., 2009). We consider three versions of COD-Cluster17, with 5k, 10k, and all data, respectively. Detailed discussion on this dataset is provided in Appendix G.

**Evaluation metrics.** We evaluate the performance of the compared approaches using a comprehensive set of metrics tailored to assess the quality of crystallization packing. These metrics include: (1) *Packing Matching (PM)* (Chisholm & Motherwell, 2005): This metric measures how well the generated molecular assemblies match the reference crystal structures in terms of spatial arrangement and packing density. Following (Liu et al., 2024c), we employ packing matching on both the atomic level (PM-atom) and the mass-center-level (PM-center) (Chisholm & Motherwell, 2005). (2) *Atomic Collision*: This follows (Cordero et al., 2008). It measures the percentage of collided atom pairs in the predicted assemblies. Atoms must maintain a minimum covalent distance governed by the balance of attractive and repulsive forces. (3) *Separation*: We extend the metric from (Xie et al., 2022; Yang et al., 2024) to our setting. A cluster of molecules is valid if the minimum distance between molecules

is above 0.5Å (Court et al., 2020). This metric is referred to as *separation* to measure the validity to avoid unphysical interactions at the molecular level. (4) *Compactness*: We propose this measure by calculating the percentage of simulated clusters where the maximum atomic pairwise distances are below 100Å. A higher compactness value suggests a more efficient arrangement, where the intermolecular spaces are minimized, leading to a denser crystalline structure. Detailed discussions on these metrics are provided in Appendix G.

**Baselines.** We compare our method with two categories of baselines: state-of-the-art deep generative models and an established domain-specific simulation tool.

*(1) Deep generative baselines.* For generative models, we evaluate our approach against *GNN-MD* (Liu et al., 2024c), *CrystalSDE* (Liu et al., 2024c), *CrystalFlow* (Liu et al., 2024c), and different variations of them, including *CrystalSDE-VE*, *CrystalSDE-VP*,

Table 2: Ablation studies of PackMol and AssembleFlow variants.

|  | PackMol | AssembleFlow-Atom | AssembleFlow-Molecule |
|---|---|---|---|
| Dataset: **COD-Cluster17-5K** | | | |
| PM (atom) ↓ | $7.10 \pm 0.05$ | $7.27 \pm 0.04$ | $7.67 \pm 0.10$ |
| PM (center) ↓ | $6.05 \pm 0.04$ | $6.13 \pm 0.10$ | $6.77 \pm 0.10$ |
| Collision ↓ | $0.32 \pm 0.00$ | $0.33 \pm 0.00$ | $0.37 \pm 0.01$ |
| Separation ↑ | $99.56 \pm 0.08$ | $97.64 \pm 0.36$ | $92.95 \pm 0.16$ |
| Dataset: **COD-Cluster17-10K** | | | |
| PM (atom) ↓ | $7.16 \pm 0.01$ | $7.38 \pm 0.03$ | $7.65 \pm 0.17$ |
| PM (center) ↓ | $6.11 \pm 0.01$ | $6.21 \pm 0.05$ | $6.69 \pm 0.20$ |
| Collision ↓ | $0.30 \pm 0.00$ | $0.31 \pm 0.00$ | $0.35 \pm 0.01$ |
| Separation ↑ | $99.45 \pm 0.10$ | $97.73 \pm 0.16$ | $92.67 \pm 0.32$ |
| Dataset: **COD-Cluster17-All** | | | |
| PM (atom) ↓ | $7.15 \pm 0.01$ | $7.37 \pm 0.01$ | $7.47 \pm 0.05$ |
| PM (center) ↓ | $6.09 \pm 0.01$ | $6.21 \pm 0.01$ | $6.36 \pm 0.06$ |
| Collision ↓ | $0.30 \pm 0.00$ | $0.31 \pm 0.00$ | $0.33 \pm 0.00$ |
| Separation ↑ | $99.42 \pm 0.03$ | $98.15 \pm 0.22$ | $95.66 \pm 0.08$ |

*CrystalFlow-VE*, *CrystalFlow-VP*, and *CrystalFlow-LERP*. These models employ various mechanisms to handle the challenges of molecular crystallization. *CrystalSDE-VE* and *CrystalSDE-VP* use stochastic differential equations to model diffusion processes under different parameterizations. *CrystalFlow-VE* and *CrystalFlow-VP* apply flow matching principles for diffusion-based interpolation path, with the latter focusing on variance-preserving methods. *CrystalFlow-LERP* utilizes linear interpolation to handle molecular transformations, striking a balance between computational complexity and performance.

*(2) Domain-specific simulation baseline.* We also compare our method with *PackMol* (Martínez et al., 2009), a well-established simulation tool widely used in the field for molecular packing. *PackMol* has long been a go-to solution for chemistry and material experts due to its ability to generate initial molecular configurations for follow-up simulations, making it an important and relevant baseline for evaluating molecular assembly tasks. More detail on this baseline is in Appendix F.

## 4.2 MAIN RESULTS

The comparison results with the generative modeling baselines and the simulation model are presented in Tables 1 and 2, respectively.

As shown in Table 1, AssembleFlow significantly outperformed all six deep generative models across almost all metrics. For example, AssembleFlow improved Packing Matching by at least 45% compared to other models. Notably, most baselines struggled with rigid packing, leading to very low Separation scores, except for CrystalFlow-LERP. For example, AssembleFlow achieved a Separation rate of 97.64%, while GNN-MD only reached 0.22%.

As shown in Table 2, when compared to the domain-specific tool PackMol, our data-driven approach demonstrates strong assembly performance relative to this widely used simulation method. Remarkably, our method achieves 100% validity, matching that of the domain-specific simulation tool. While this outcome highlights the promise of AssembleFlow, it is expected for PackMol, as it leverages well-established domain knowledge and heuristic physical rules to determine molecular orientations. Promisingly, both methods achieved a very low Collision rate. The results indicate that the data-driven AssembleFlow performs comparably to the domain simulation tool PackMol.

## 4.3 ABLATION STUDIES

**Velocity function.** As discussed in Section 3.5, AssembleFlow can utilize two types of SE(3)-equivariant velocity functions. We here evaluate their performance and, as shown in the last two columns of Table 2, AssembleFlow-Atom generally outperforms AssembleFlow-Molecule, as atomic-level information provides more detailed

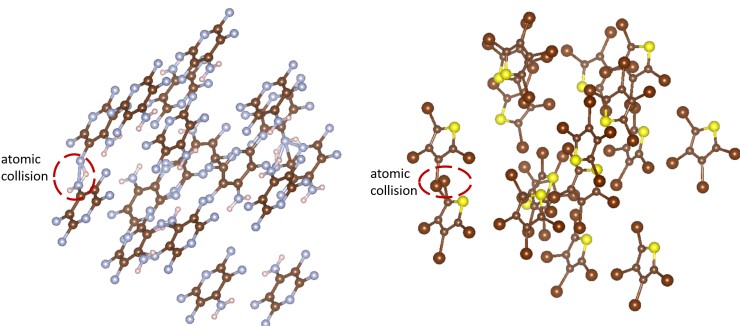

Figure 4: Atomic collisions (red circles) in predicted assemblies.

geometric insights. Despite this, both variations of AssembleFlow exhibit strong results, significantly surpassing other deep learning baselines.

**Atomic collision.** To show atomic collisions of the assemblies, we visualize, in Figure 4, two atomic collisions in assemblies predicted by AssembleFlow, where two molecules collide into each other (indicated by the red circle).

**Computational cost.** In Figure 5, we present the computation time needed for AssembleFlow and PackMol. It reveals that our data-driven method achieves a 25-fold reduction in computational costs. This suggests that our method can be scaled effectively for larger molecular systems and datasets.

## 5 CONCLUSION AND OUTLOOK

We introduced AssembleFlow, a generative model that maintains the inherent rigidity of molecular structures during assembly. By using inertial frames for positional references at the molecular level, AssembleFlow accurately tracks molecular orientation and motion. It decomposes transformations into translations and rotations, enforcing rigidity throughout the generation process. This innovative approach enables the model to separately learn velocity functions using linear and spherical interpolation for accurate rigid molecular assembly.

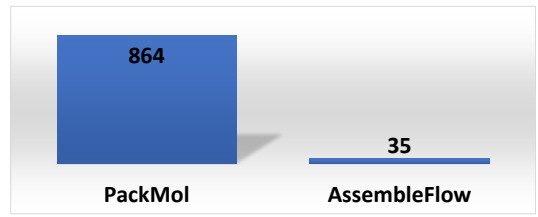

Figure 5: Comparison of computational cost in hours for 10,543 molecule clusters from COD-Cluster17. Pack-Mol requires around 864 hours, while AssembleFlow requires 35 hours.

Empirical results on COD-Cluster17 show that AssembleFlow outperforms six state-of-the-art deep learning models while maintaining molecular integrity and achieves comparable assembly performance to an established simulation tool, significantly reducing computational costs. This suggests that AssembleFlow offers an efficient solution for large-scale molecular simulations and opens avenues for faster material generation in molecular engineering.

To the best of our knowledge, AssembleFlow is the first to implement rigid generation in SE(3) space for molecular assembly. It has the potential to be generalized to more complex and challenging scenarios, such as simulating the crystallization process of polymorphs with diverse configurations and structures. These scenarios are critical for accurately modeling real-world materials, where different polymorphic forms can exhibit vastly different properties. However, the current dataset, COD-Cluster17, lacks the necessary information to fully explore this aspect. As such, extending AssembleFlow to handle these intricate processes remains an exciting direction for future research, where more specialized datasets could enable deeper insights into material formation and polymorph behavior. We aim to address these challenges in future work.

ACKNOWLEDGMENTS

We thank the anonymous reviewers for their valuable suggestions and feedback. YB acknowledges support from NRC AI4D, CIFAR, and the CIFAR AI Chair program. This project's computational resources are provided by NRC and the Digital Research Alliance of Canada.

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

## A    RELATED WORKS

### A.1    GEOMETRIC MODELING

Geometric modeling of molecules has predominantly been applied in 3D Euclidean space (Smidt et al., 2018). It requires that the representation and generation function over the molecular system remain equivariant to the rotations and translations, *i.e.*, the SE(3)-equivariance, ensuring that molecular properties are preserved regardless of orientation or position in space (Smidt et al., 2018; Zhang et al., 2023; Liu et al., 2023).

We note that reflection (or chirality) is also an important factor in geometric molecule modeling. For multi-component molecular systems like protein-ligand binding complexes, each individual molecule can lead to different energies and corresponding properties with distinct chiralities (Brooks et al., 2011; McVicker & O'Boyle, 2024). Also, as shown in AlphaFold2, the natural molecules should be sensitive to the chirality (Jumper et al., 2021). Thus, more physically accurate geometric modeling should be SE(3)-equivariant and reflection-antisymmetric, and we have shown how our proposed AssembleFlow satisfies this in Section 3.

We also want to highlight another line of research that downplays the importance of symmetry in molecular modeling, as demonstrated by models like AlphaFold3 (Abramson et al., 2024) and XYZTransformer (Flam-Shepherd & Aspuru-Guzik, 2023). These models avoid incorporating geometric symmetries because enforcing group symmetry constraints, such as SE(3)-equivariance, can limit a model's expressiveness. In some domain-specific tasks, relaxing these constraints has resulted in strong performance. However, in the case of crystallization, maintaining molecular rigidity—a key symmetric property—is crucial. As demonstrated in previous work (Liu et al., 2024c), neglecting these equivariance constraints during molecular crystallization leads to unrealistic molecular structures.

### A.2    GENERATIVE MODELS ON GEOMETRIC DATA

The geometric modeling on the continuous 3D Euclidean space can be naturally merged with deep generative frameworks, where the goal is to learn the geometric data distribution and generate new molecules. The deep generative models include but are not limited to denoising score matching (Vincent, 2011; Song & Ermon, 2019), denoising diffusion probabilistic model (Ho et al., 2020), and flow matching (Lipman et al., 2022; Liu et al., 2022; Albergo & Vanden-Eijnden, 2022). Such geometric generative models have been widely adopted for molecule and material generation (Hoogeboom et al., 2022; Jiao et al., 2024; Luo et al., 2024), structure-based drug design (Guan et al., 2023; 2022; Liu et al., 2024b), and molecular dynamics simulation (Doerr et al., 2021; Arts et al., 2023; Liu et al., 2024a).

### A.3    GENERATIVE MODELS ON RIGID GEOMETRIC DATA

Though our work is the first to apply rigid generation in SE(3) space for molecular packing/crystallization, similar ideas have been carried out for protein backbone generation (Jumper et al., 2021; Yim et al., 2023a;b; Bose et al., 2023; Huguet et al., 2024) and relevant works (Köhler et al., 2023; Ganea et al., 2021; Klein et al., 2024)) In the protein backbone generation and folding setting, there exists a well-defined local frame structure for each residue: the backbone atom pairs $(C - N)$ and $(C_\alpha - C)$ form two bases, and their cross product leads to the third basis, which is a normal vector perpendicular to the two bases. Thus, the rotation matrix can be easily constructed based on such a local frame structure. Such a modeling paradigm is appealing for macromolecules like proteins to reduce the computational cost.

In this field, AlphaFold2 adopts this local-frame idea to model the folding process (Jumper et al., 2021), while FrameDiff applies this idea and denoising diffusion model for protein structure generation (Yim et al., 2023b). Similarly, FrameFlow and FoldFlow integrate local frames with flow matching to learn the protein dynamics (Yim et al., 2023a; Bose et al., 2023; Huguet et al., 2024). However, this approach cannot be easily extended to crystallization tasks, as constructing reliable local frames to establish positional references for assembling molecules is not straightforward. Instead, we innovatively introduce inertial frames to achieve this goal.

### A.4 COMPARISON BETWEEN SPHERICAL INTERPOLATION AND EXPONENTIAL MAP INTERPOLATION ON SO(3)

In addition to spherical interpolation, we could also consider the exponential map interpolation, as used in (Riemannian FM, FrameFlow, and FoldFlow). In this section, we would like to compare the theoretical differences. We further conduct experiments for empirical comparison, and please check Appendix H.2 for more details.

We mark exponential map interpolation as (EMLERP), and it is defined as:

$$\text{EMLERP}(\boldsymbol{r}_0, \boldsymbol{r}_1, t) = \exp_{\boldsymbol{r}_0}(t \log_{\boldsymbol{r}_0}(\boldsymbol{r}_1)). \tag{15}$$

We then discuss how FoldFlow and FoldFlow2 utilized this equation since they are the latest work along this line of using EMLERP. By utilizing the tangent space of the manifold and axis-angle representation, existing works (FoldFlow, FoldFlow2) have been using an approximated closed-form solution for the derivative:

$$\frac{d}{dt}\text{EMLERP}(\boldsymbol{r}_0, \boldsymbol{r}_1, t) = \log_{\boldsymbol{r}_t} \frac{\boldsymbol{r}_0}{t}. \tag{16}$$

Thus, their objective function on SO(3) is:

$$\begin{aligned}
\mathcal{L}_{\text{SO}(3)} &= \|\frac{d}{dt}\text{EMLERP}(\boldsymbol{r}_0, \boldsymbol{r}_1, t) - \boldsymbol{v}_{\theta,\text{SO}(3)}(\boldsymbol{x}_t, \boldsymbol{r}_t, t)\|^2 \\
&= \|\log_{\boldsymbol{r}_t} \frac{\boldsymbol{r}_0}{t} - \boldsymbol{v}_{\theta,\text{SO}(3)}(\boldsymbol{x}_t, \boldsymbol{r}_t, t)\|^2.
\end{aligned} \tag{17}$$

To sum up, these two methods do not have a clear methodological advantage over each other; however, the EMLERP considers more approximation tricks in implementation. We summarize the main differences in Table 3.

Table 3: Comparison between the interpolation paths in AssembleFlow and FrameFlow.

|  | AssembleFlow (ours) | FoldFlow |
|---|---|---|
| Reference Frame | Eigenvectors of inertial frames | Gram-Schmidt on $N$-$C$-$C_\alpha$ |
| SO(3) Interpolation | Spherical Interpolation | Exponential Map Interpolation |
| Equation | $\frac{\sin((1-t)\omega)\boldsymbol{q}_0 + \sin(t\omega)\boldsymbol{q}_1}{\sin(\omega)}$ | $\exp_{\boldsymbol{r}_0}(t \log_{\boldsymbol{r}_0}(\boldsymbol{r}_1))$ |
| Derivative | Equation (9) | Equation (16) |
| Rotation Representation (for Velocity / Objective Function) | Quaternion | Rotation Matrix or Axis-angle |
| Reparameterization | Yes | No |

### A.5 COMPARISON BETWEEN OUR WORK AND RIGID STRUCTURE SAMPLING (KÖHLER ET AL., 2023)

We would like to emphasize that our work differs from (Köhler et al., 2023) in the following fundamental aspects. Noticeably, we would like to emphasize that our work can be seen as an extension of (Köhler et al., 2023), yet addressing more practical and challenging problems, including rigid modeling on arbitrary molecules, SE(3)-equivariant modeling, and interpolation modeling.

1. **Experiment and Data Difference**: (Köhler et al., 2023) targets at modeling the transition, e.g., water molecules at different temperatures (no code or comments related to methane rigid modeling were found in the GitHub repository). The MD simulation can provide samples at the stable or equilibrium status. Our work models the transition from unstable to stable conformations.

2. **Dynamic Transition Modeling vs. Stable Structure Sampling**: Unlike (Köhler et al., 2023) which focuses on stable structure sampling, our work models the dynamic transition process from weakly correlated (unstable) structures to strongly correlated (stable) structures. Notably, dynamic transition modeling toward stability is identified as **a nontrivial next step in the ICML work (Köhler et al., 2023)**, which states '... a flow model for the positions that can handle ... phase transitions'. We also want to emphasize that the transition of a molecular cluster from weakly correlated (unstable) structures to strongly correlated (stable) structures is a special case of the general dynamics. The limitation here is the data insufficiency (lack of intermediate snapshots), not modeling.

3. **Objective Difference**: (Köhler et al., 2023) first introduces molecular equilibrium sampling with Boltzmann distribution in Eq 1,2. But this is not the goal, (Köhler et al., 2023) changes to estimating log-ratio: $\nabla F = -\log(Z_{\alpha_1}/Z_{\alpha_0})$ between two configurations in Eq 3. This measure estimates the energy difference and tells which state is more stable. (Köhler et al., 2023) says using trackable priors like Gaussian can be biased, thus it takes the insight from previous work, and targets solving the problems Eq 3 and 7 (learned free energy perturbation). (Notice that other equations Eq 2,4,5 are preliminaries, not directly related to the core method in this work.) This reveals another theoretical difference between this work and our work: (Köhler et al., 2023) is estimating the upper bound of log-ratio between two stable states (with MD simulations), and ours is directly modeling $p(X_1|X_0)$ from unstable to stable transition.

4. **Use of Inertial Frames for Rigid Modeling**: (Köhler et al., 2023) specifically models the frame for H-O-H (codes here) (no code or comments related to methane rigid modeling were found in the GitHub repository). In contrast, our approach is more generalizable, as the inertial frame can serve as a reference frame for any molecule. Thus, (Köhler et al., 2023) cannot be directly applied to our dataset, as it is limited to few types of constructed frames, and our work can be viewed as an extension of (Köhler et al., 2023) to a more general setting.

5. **SE(3)-equivariant Symmetry Modeling**: (Köhler et al., 2023) states that it has a limitation on not 'exploiting the SE(3) symmetry of jointly moving all rigid bodies'. We solve this issue by introducing the SE(3)-equivariant modeling from two granularities.

6. **Limitations of Coupling Layers in Normalizing Flow**: (Köhler et al., 2023) does modeling with an extra bijectivity constraint in coupling layers, limiting the model capacity (Ho et al., 2019). Flow matching enables flexible velocity functions under the interpolation framework.

We list the comparison in the table below:

Table 4: Comparison between (Köhler et al., 2023) and AssembleFlow.

| | Paper (Köhler et al., 2023) | AssembleFlow (ours) |
|---|---|---|
| Experiments | Transition between two stable conformations | Transition from unstable to stable |
| Data | Water and methane (experiment missing on GitHub repo) molecules | COD organic molecules |
| Frame construction | H-O-H frame specific for water molecule (No code or comments related to methane rigid modeling were found in the GitHub repository) | Inertial frame for any organic molecule |
| Objective function | Upper bound of log-ratio between two stable conformations for energy difference | Direct estimation of conditional density from unstable to stable |
| Modeling SE(3) symmetry in moving rigid bodies | No | Yes |
| Avoid bijectivity constraint in coupling layers | No | Yes |

## B  ROTATION REPRESENTATION

In this section, we will be mainly discussing three types of rotation representations. It is important to note that "representation" here refers to the data structure commonly used in the machine learning community, rather than the concept of a representation space.

In Appendix C.1, we will explain how to use inertials to represent the rigid structures of molecules in a cluster. With such a rigid representation, we can then decompose $SE(3)$ transformation into a tuple of $SO(3)$ and $\mathbb{R}^3$ transformations.

A natural way to represent the $SO(3)$ transformation is by using the rotation matrix, as will be introduced in Appendix B.1. However the rotation matrix is not flexible and it must satisfy specific mathematical properties to make sure it is a valid rotation in space, so we need a more flexible and efficient representation. To this end, we would like to introduce axis-angle representation in Appendix B.2 and quaternion representation in Appendix B.3. The axis-angle representation and quaternion representation are closely related, and their transformation is discussed in Appendix B.4. Last but not least, the transformation between the rotation matrix and quaternion is in Appendix B.5, and the transformation between axis-angle representation and rotation matrix representation is in Appendix B.6. An overview of the rotation representation and the corresponding transformations are listed in Figure 6.

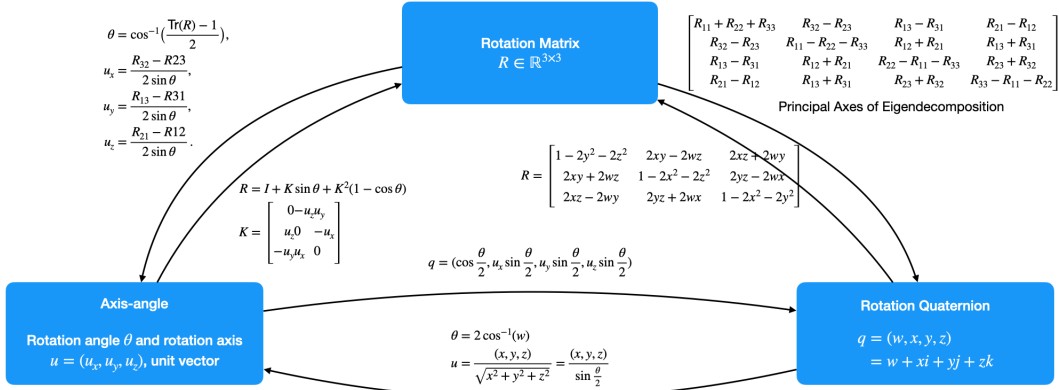

Figure 6: Transformation between rotation matrix, quaternion, and axis-angle representation.

### B.1  ROTATION REPRESENTATION WITH ROTATION MATRIX

**Definition**   Rotation matrix is $R \in \mathbb{R}^{3\times3}$, satisfying two conditions:

1. Orthogonality: The rows and columns of $R$ are orthonormal vectors. $R^T R = R R^T = I$.
2. Determinant: The determinant of $R$ must be 1. $\det(R) = 1$.

The set of all orthogonal matrices of size 3 with determinant 1 is a representation of a group known as the special orthogonal group $SO(3)$. To generate a rotation matrix in $SO(3)$, certain properties for the rotation matrix must be satisfied, and such constrained generation is challenging. Thus, an alternative way of representing the rotation matrix is preferred. To this end, we consider axis-angle representation, as described below.

**Rotation with Rotation Matrix**   To rotation a point, $p = (x, y, z)$, in the 3D Euclidean space, the rotation operation is:

$$
\begin{bmatrix} x' \\ y' \\ z' \end{bmatrix} = R \begin{bmatrix} x \\ y \\ z \end{bmatrix} = \begin{bmatrix} R_{11} & R_{12} & R_{13} \\ R_{21} & R_{22} & R_{23} \\ R_{31} & R_{32} & R_{33} \end{bmatrix} \begin{bmatrix} x \\ y \\ z \end{bmatrix}. \tag{18}
$$

**Properties**

- If we have multiple rotation matrices, and we want to yield a single matrix that combines these rotations into one, then we have two options:

- Extrinsic rotations. All rotations refer to a fixed and global coordinate system, and the rotation matrices are ordered from the right to the left. If we apply rotations $R_1$, $R_2$, and $R_3$, then it is written as $R = R_3 R_2 R_1$.
- Intrinsic rotations. A rotation refers to the last rotated coordinate system, and the rotation matrices are ordered from the left to the right. If we apply rotations $R_1$, $R_2$, and $R_3$, then it is written as $R = R_1 R_2 R_3$.

### B.2 ROTATION REPRESENTATION WITH AXIS-ANGLE REPRESENTATION

The axis-angle represents the rotation by its angle $\theta$ and the rotation axis $\boldsymbol{u}$. For example, a rotation of 180 degrees around the Y-Axis would be represented as $\theta = 180$, $\boldsymbol{u} = (0, 1, 0)$. The representation is very intuitive, but for actually applying the rotation, another representation is required, such as a quaternion or rotation matrix.

**Definition** The axis-angle representation of a rotation is defined by two components:

1. Rotation axis: a unit vector $\boldsymbol{u} = (u_x, u_y, u_z)$ that specifies the direction of rotation, $\|\boldsymbol{u}\| = 1$.
2. A scalar $\theta$ specifies the angle of rotation around the rotation axis.

For annotation, an axis-angle rotation can thus be presented by four numbers as $(\theta, \hat{x}, \hat{y}, \hat{z})$.

### B.3 ROTATION REPRESENTATION WITH QUATERNION REPRESENTATION

Quaternion represents a rotation by a 4D vector and it is a more concise representation than a rotation matrix. It requires more math and is less intuitive, but is a much more powerful representation. Quaternion representation has been widely used in rigid motion modeling, robotics modeling, and quantum mechanics (*e.g.*, the spin of an electron and the polarization of a photon). In this work, we are focusing on the first case, rigid motion modeling.

**Definition** A quaternion $\boldsymbol{q}$ is:

$$\boldsymbol{q} = w + xi + yj + zk = (w, \boldsymbol{v}), \tag{19}$$

where $w, x, y, z$ are real numbers, and $i, j, k$ are the fundamental quaternion units. The $w$ is the real part, and $\boldsymbol{v} = (x, y, z)$ is a 3D vector representing the imaginary part.

**Multipliation of basis elements** The multiplication for the basis elements $i, j, k$ is defined below:

$$\begin{aligned}
i^2 = j^2 &= k^2 = -1 \\
ij = -ji &= k \\
jk = -kj &= i \\
ki = -ik &= j \\
ijk &= -1.
\end{aligned} \tag{20}$$

**Quanternion multiplication: Hamilton product** This can give us the quaternion multiplication, a.k.a., Hamilton product. For two quaternions $\boldsymbol{r} = (r_0, r_1, r_2, r_3)$ and $\boldsymbol{s} = (s_0, s_1, s_2, s_3)$:

$$\boldsymbol{t} = \boldsymbol{r}\boldsymbol{s}, \tag{21}$$

where

$$\begin{aligned}
t_0 &= r_0 s_0 - r_1 s_1 - r_2 s_2 - r_3 s_3 \\
t_1 &= r_0 s_1 + r_1 s_0 - r_2 s_3 + r_3 s_2 \\
t_2 &= r_0 s_2 + r_1 s_3 + r_2 s_0 - r_3 s_1 \\
t_3 &= r_0 s_3 - r_1 s_2 + r_2 s_1 + r_3 s_0.
\end{aligned} \tag{22}$$

**Point rotation with quaternion** We rotate a point $\boldsymbol{v} = (v_x, v_y, v_z)$ by the quaternion $\boldsymbol{q} = (w, x, y, z)$ using the following three steps:

1. Transform $\boldsymbol{v}$ to quaternion $\boldsymbol{p} = (0, v_x, v_y, v_z)$.

2. Construct the conjugate quaternion $\boldsymbol{q}^* = (w, -x, -y, -z)$.
3. There are two types of rotation operations:
   (a) Active rotation: $\boldsymbol{p}' = \boldsymbol{q}^*\boldsymbol{p}\boldsymbol{q}$, when the point is rotated w.r.t. the coordinate system.
   (b) Passive rotation: $\boldsymbol{p}' = \boldsymbol{q}\boldsymbol{p}\boldsymbol{q}^*$, when the coordinate system is rotated w.r.t. the point.
   Notice that the two rotations are opposite from each other. In our case, we use the passive rotation.
4. The resulting vector $\boldsymbol{v}'$ is extracted from the imaginary part of $\boldsymbol{p}'$.

**Spherical Interpolation (SLPER) for Quaternion Interpolation**  Quaternions are often preferred for interpolating between rotations because they offer smoother interpolation than axis-angle representation. The spherical interpolation defines the geodesic over the rotation group.

$$\text{SLERP}(\boldsymbol{q}_0, \boldsymbol{q}_1, t) = \frac{\sin((1-t)\omega)\boldsymbol{q}_0 + \sin(t\omega)\boldsymbol{q}_1}{\sin(\omega)}, \tag{23}$$

where $\omega$ is the angle between $\boldsymbol{q}_0$ and $\boldsymbol{q}_1$.

**Properties**

- A quaternion is a unit quaternion if $\|\boldsymbol{q}\| = w^2 + x^2 + y^2 + z^2 = 1$.
- All rotation quaternions must be unit quaternions.
- A rotation of $\boldsymbol{q}_a$ followed by a rotation of $\boldsymbol{q}_b$ can be combined into a single rotation: $\boldsymbol{q}_c = \boldsymbol{q}_b\boldsymbol{q}_a$. Note that order matters.
- The conjugate of a quaternion is $\boldsymbol{q}^* = (w, -x, -y, -z)$.
- The inverse of a rotation quaternion is $\boldsymbol{q}^{-1} = \boldsymbol{q}^*$. Then we can see that $\boldsymbol{q}\boldsymbol{q}^{-1} = \boldsymbol{q}\boldsymbol{q}^* = (1, 0, 0, 0)$.
- Quaternion multiplication is associative: $\boldsymbol{abc} = \boldsymbol{a}(\boldsymbol{bc})$.
- Quaternion multiplication is not commutative: $\boldsymbol{ab} \neq \boldsymbol{ba}$.

### B.4 TRANSFORMATION BETWEEN AXIS-ANGLE AND QUATERNION

**Axis-angle representation to quaternion representation**  Axis-angle representation is $\boldsymbol{u} = (u_x, u_y, u_z)$ equipped with a rotation angle $\theta$, and the rotation quaternion is unitary, *i.e.*, $w^2 + x^2 + y^2 + z^2 = 1$. The quaternion is thus:

$$\boldsymbol{q} = \cos\left(\frac{\theta}{2}\right) + \sin\left(\frac{\theta}{2}\right)(u_x i + u_y j + u_z k), \tag{24}$$

or equivalently in the vector form:

$$\boldsymbol{q} = \left(\cos\left(\frac{\theta}{2}\right), \sin\left(\frac{\theta}{2}\right)u_x, \sin\left(\frac{\theta}{2}\right)u_y, \sin\left(\frac{\theta}{2}\right)u_z\right). \tag{25}$$

**Quaternion representation to axis-angle representation**  Quaternion is $\boldsymbol{q} = (w, x, y, z)$, and to convert it to axis-angle representation:

1. Compute the angle $\theta = 2\cos^{-1}(w)$.
2. Compute the axis $\boldsymbol{u}$:

$$\boldsymbol{u} = \frac{(x, y, z)}{\sqrt{x^2 + y^2 + z^2}} = \frac{(x, y, z)}{\sin\frac{\theta}{2}}. \tag{26}$$

### B.5 TRANSFORMATION BETWEEN QUATERNION AND ROTATION

**Quaternion to rotation matrix**  Given a quaternion $\boldsymbol{q} = (w, x, y, z)$, the corresponding rotation matrix is

$$R = \begin{bmatrix} 1 - 2y^2 - 2z^2 & 2xy - 2wz & 2xz + 2wy \\ 2xy + 2wz & 1 - 2x^2 - 2z^2 & 2yz - 2wx \\ 2xz - 2wy & 2yz + 2wx & 1 - 2x^2 - 2y^2 \end{bmatrix} \tag{27}$$

**Rotation matrix to rotation quaternion**  Given a rotation matrix $R = \begin{bmatrix} R_{11} & R_{12} & R_{13} \\ R_{21} & R_{22} & R_{23} \\ R_{31} & R_{32} & R_{33} \end{bmatrix}$, then

we first calculate the magnitude of four quaternion components as below:

$$
\begin{aligned}
|q_0| &= \sqrt{\frac{1 + R_{11} + R_{22} + R_{33}}{4}} \\
|q_1| &= \sqrt{\frac{1 + R_{11} - R_{22} - R_{33}}{4}} \\
|q_2| &= \sqrt{\frac{1 - R_{11} + R_{22} - R_{33}}{4}} \\
|q_3| &= \sqrt{\frac{1 - R_{11} - R_{22} + R_{33}}{4}}
\end{aligned}
\tag{28}
$$

To find the signs of the four elements, we can find the largest magnitude:

- If $|q_0|$ is the largest, then

$$
w = q_0, \qquad x = \frac{R_{32} - R_{23}}{4w}, \qquad y = \frac{R_{13} - R_{31}}{4w}, \qquad z = \frac{R_{21} - R_{12}}{4w}.
\tag{29}
$$

- If $|q_1|$ is the largest, then

$$
x = q_1, \qquad w = \frac{R_{32} - R_{23}}{4x}, \qquad y = \frac{R_{12} + R_{21}}{4x}, \qquad z = \frac{R_{13} + R_{31}}{4x}.
\tag{30}
$$

- If $|q_2|$ is the largest, then

$$
y = q_2, \qquad w = \frac{R_{13} - R_{31}}{4y}, \qquad x = \frac{R_{12} + R_{21}}{4y}, \qquad z = \frac{R_{23} + R_{32}}{4y}.
\tag{31}
$$

- If $|q_3|$ is the largest, then

$$
z = q_3, \qquad w = \frac{R_{21} - R_{12}}{4z}, \qquad x = \frac{R_{13} + R_{31}}{4z}, \qquad y = \frac{R_{23} + R_{32}}{4z}.
\tag{32}
$$

The sign is ambiguous because any rotation has two possible quaternion representations. If one is known, the other one can be found by taking the negative of all four terms.

Besides, there exist other solutions, for instance, extracting quaternion from rotation matrix using eigendecomposition (Horn, 1987; Bar-Itzhack, 2000). We first construct a matrix $K$ with:

$$
K = \frac{1}{3} \begin{bmatrix} R_{11} + R_{22} + R_{33} & R_{32} - R_{23} & R_{13} - R_{31} & R_{21} - R_{12} \\ R_{32} - R_{23} & R_{11} - R_{22} - R_{33} & R_{12} + R_{21} & R_{13} + R_{31} \\ R_{13} - R_{31} & R_{12} + R_{21} & R_{22} - R_{11} - R_{33} & R_{23} + R_{32} \\ R_{21} - R_{12} & R_{13} + R_{31} & R_{23} + R_{32} & R_{33} - R_{11} - R_{22} \end{bmatrix}.
\tag{33}
$$

Then we perform eigendecomposition $K = V \Lambda V^T$, where $\Lambda$ is a diagonal matrix with eigenvalues and $V$ is the matrix with eigenvectors as columns. Finally, we pick up the eigenvector w.r.t. the largest eigenvalue, and this eigenvector is the unit quaternion.

### B.6 TRANSFORMATION BETWEEN AXIS-ANGLE REPRESENTATION AND ROTATION MATRIX

**Axis-angle representation to rotation matrix**  Construct the skew-symmetric matrix:

$$
K = \begin{bmatrix} 0 & -u_z & u_y \\ u_z & 0 & -u_x \\ -u_y & u_x & 0 \end{bmatrix}
\tag{34}
$$

According to the Rodrigue's rotation formula, we have:

$$
R = I + \sin(\theta)K + (1 - \cos(\theta))K^2.
\tag{35}
$$

**Rotation matrix to axis-angle representation**  The rotation angle can be computed using the trace of $R$, *i.e.*,

$$\theta = \cos^{-1}\Big(\frac{\text{Tr}(R) - 1}{2}\Big),$$
(36)

where $\text{Tr}(R)$ is the trace, as $\text{Tr}(R) = R_{11} + R_{22} + R_{33}$.

Then we calculate the rotation direction or rotation axis $\boldsymbol{u} = (u_x, u_y, u_z)$, as

$$u_x = \frac{R_{32} - R_{23}}{2\sin\theta}, \qquad u_y = \frac{R_{13} - R_{31}}{2\sin\theta}, \qquad u_z = \frac{R_{21} - R_{12}}{2\sin\theta}.$$
(37)

Notice that when $\theta = 0$, there is no definition of rotation axis and the whole rotation matrix $R$ is unitary. When $\theta = \pi$, because $\sin\theta = 0$, then we need other methods (*e.g.*, eigendecomposition) to determine the rotation axis.

## C  INERTIAL FRAME FOR RIGID BODY

### C.1  RIGID REPRESENTATION OF MOLECULE

We employ the following four sequential steps to derive the reference frames that construct the rotation matrix from $N$ atomic positions $r$:

- Calculate the mass center: $c = \frac{1}{N} \sum_i r_i$.
- Adjust position relative to the center $r_i = r_i - c$.
- Compute the inertia tensor $\hat{I} = \sum_i \|r_i\|^2 I - r_i r_i^T$, where $I$ is the unit diagonal matrix.
- Obtain the principal axes of inertia by applying eigen-decomposition on $\hat{I}$. We have $\hat{I} = Q \Lambda Q^T$, where $Q$ is the orthogonal matrix whose columns are the eigenvectors of $\hat{I}$, and $\Lambda$ is the diagonal matrix whose elements are the eigenvalues $\lambda_i$ of $\hat{I}$, representing the principal moments of inertial along the principal axes.

### C.2  ORTHOGONAL MATRIX

In linear algebra, an orthogonal matrix or orthonormal matrix is a square matrix whose columns and rows are orthonormal vectors. This can be written as

$$Q^T Q = QQ^T = I. \tag{38}$$

This leads to the equivalent characterization: a matrix $Q$ is orthogonal if its transpose is equal to its inverse:

$$Q^T = Q^{-1}. \tag{39}$$

Notice that when discussing matrices, the two terms (orthogonal and orthonormal) can be used interchangeably.

If $Q$ is a square matrix, then the conditions $RR^T = I$ and $R^T R = I$ are equivalent. Proof sketch: $R^T R = I$ and $R^T R R^{-1} = R^{-1}$, so $R^T = R^{-1}$. This can give us $RR^T = I$.

### C.3  ROTATION MATRIX FROM RIGID BODY

From Appendix C.1, we can construct the inertial tensors. Then we employ eigenvalue decomposition on the inertial tensor. The normalized eigenvectors $v_1, v_2, v_3$ form an orthonormal basis, which can be used to construct the rotation matrix, *i.e.*,

$$R = \begin{bmatrix} v_1 & v_2 & v_3 \end{bmatrix}. \tag{40}$$

**Eigendecomposition of Inertial Tensors**  For inertial tensor $I$, the decomposition is: with $I v_i = \lambda_i v_i$, where $\lambda_i$ are eigenvalues and $v_i \in \mathbb{R}^{3 \times 1}$ are eigenvectors. Thus, we can have

$$\begin{aligned} I v &= \lambda v \\ I R &= R \Lambda \\ I &= R \Lambda R^{-1}, \end{aligned} \tag{41}$$

where $\Lambda = \begin{bmatrix} \lambda_0 & 0 & 0 \\ 0 & \lambda_1 & 0 \\ 0 & 0 & \lambda_2 \end{bmatrix}$. Because the inertial tensors are symmetric matrices, we have that matrices $R$ are orthonormal matrices.

### C.4  EXPLORATIONS ON OTHER REFERENCE FRAME OPTION

One critical question is in addition to the Inertial frame, do we have other options for modeling the rigidity? One simple solution is to directly apply the eigendecomposition as principal component analysis (PCA) on the point clouds of centered molecules.

First, we would like to clarify that there are two roles and one important property of the inertial frame and its eigenvectors:

- Role 1: The three bases in inertial frames act as a reference or a canonical pose.
- Role 2: The three bases enable modeling the velocity function in SO3 space.
- What's more important, we expect the three bases to be numerically stable.

Then getting back to the question, though both can help build up the reference frame or canonical pose, there are certain aspects we would like to emphasize when comparing PCA and Inertial frames.

- **Canonical pose**: The key question is defining a canonical pose. If we just do PCA on the point clouds, this cannot guarantee the group symmetry in SE(3). However, if we remove the center point, then the group symmetry can be guaranteed; then we can use SVD to get the three principal components. Till this step, one can find this is somewhat similar to the inertial frame construction (Sec 3.1).
- **Difference between PCA and Inertial Frame as reference frames**: Though both can be used for building up the reference frame or canonical poses, SVD (for PCA) on the set of point clouds $N \times 3$ (N is the number of atoms) can be less numerically unstable, while eigendecomposition Inertial Frame $3 \times 3$ can be more numerically stable. We conduct an experiment to verify this.
  - Experiment setup: Suppose we have weakly-correlated structures $X_0$ and strongly-correlated structures $X_1$, and we find the corresponding bases using either eigendecomposition on centered $X$ or inertial frame $I$. The two bases are marked as $B_0 \in \mathbb{R}^{3 \times 3}$ and $B_1 \in \mathbb{R}^{3 \times 3}$.
  - Objective: We can obtain the rotation matrix with $R^T = B_0^T B_1$, and then we can rotate the whole molecular system as $\tilde{X}_1 = X_0 R^T$, and we are measuring the reconstruction errors as $\text{MSE}(X_1 - \tilde{X}_1)$. We mark the MSE using inertial frame and PCA as $\delta_{\text{Inertial}}$ and $\delta_{\text{PCA}}$, respectively. If $\delta_{\text{Inertial}} < \delta_{\text{PCA}}$, then we can conclude that using the inertial frame is more stable than using PCA, and vice versa. Notice that since the MSE reconstruction is meaningless when it is too small, so we only compare these two frames when at least one of them has reconstruction greater than or equal to a threshold $\theta$.
  - Results: The comparison results are in Table 5, and we can observe that in general, using the inertial frame is more stable than PCA. We are listing multiple reconstruction threshold $\theta$ in Table 5, and we are using $\theta = 1e - 3$ in the main article.

Table 5: Comparison of using inertial frame and PCA for reconstruction (%).

| $\theta$ | 1e-3 | 1e-4 | 1e-5 | 1e-6 | 1e-7 | 1e-8 |
|---|---|---|---|---|---|---|
| $P(\delta_{\text{Inertial}} < \delta_{\text{PCA}})$ | 0.434 | 0.884 | 1.338 | 1.794 | 2.254 | 2.727 |
| $P(\delta_{\text{Inertial}} > \delta_{\text{PCA}})$ | 0.371 | 0.756 | 1.147 | 1.539 | 1.934 | 2.345 |

## C.5 KABSCH ALGORITHM

Kabsch algorithm is one way to compute the optimal rotation matrix that minimizes the root-mean-square deviation (RMSD) between two sets of points (atoms in our case). However, it is guaranteed in the COD-Cluster17 dataset that the molecules in weakly correlated structures can rotate to molecules in strongly correlated structures; in other words, the RMSD can be approximately 0 if we use the Kabsch algorithm, which is equivalent to calculating the rotation matrix directly after we fix the poses. We have shown how to calculate the rotation matrix in the experiment above.

## D    PROOF OF THEOREM 1

*Proof.* For three vectors, we can easily find a counter-example, as illustrated in Figure 3 (a, b). Figure 3 (a, b) describes two cases where we have the same initial frame, and we can rotate it to two different final frames with two rotation matrices, yet the righthandness still matches. We can easily see that there are four options of rotation matrices in this case, and we cannot uniquely determine the final inertial frame in this case.

More rigorously, let us first assume that there exists a rotation transformation $R$ that can transform the initial coordinate system $Q_i$ to the final coordinate system $Q_f$, as:

$$\begin{bmatrix} Q_{f,0} \\ Q_{f,1} \\ Q_{f,2} \end{bmatrix}^T = R \cdot \begin{bmatrix} Q_{i,0} \\ Q_{i,1} \\ Q_{i,2} \end{bmatrix}^T \tag{42}$$

First, as Lemma 1, we should change either zero or two directions for direction alignment.

Then without loss of generality, we can assume the two directions to be the last two axes. Thus, we can obtain a rotation matrix $R'$ such that $R'$ is rotating $R$ along vector $Q_{f,0}$ with 180 degrees. We can represent $R'$ using Rodrigue's rotation formula, as $R' = (2Q_{f,0}Q_{f,0}^T - I)R$. Thus, we can have:

$$R' \cdot \begin{bmatrix} Q_{i,0} \\ Q_{i,1} \\ Q_{i,2} \end{bmatrix}^T = (2Q_{f,0}Q_{f,0}^T - I) \begin{bmatrix} Q_{f,0} \\ Q_{f,1} \\ Q_{f,2} \end{bmatrix}^T = \begin{bmatrix} Q_{f,0} \\ -Q_{f,1} \\ -Q_{f,2} \end{bmatrix}^T \tag{43}$$

This is essentially saying starting from one initial frame, we can have multiple matched final frames. Thus, using only three vectors cannot uniquely determine the direction matching. We provide two examples in Figure 3 (a, b).

For the four vectors, we introduce an extra atom into the inertial frame system, and such an extra atom point is nonplanar to the three base axes. Then the problem becomes: starting from an initial frame and an extra point, can we find multiple rotation matrices such that the final frames have reflected directions? To be more rigorous, let us have the following formulation.

First, let us assume we have this rotation matrix:

$$\begin{bmatrix} Q_{f,0} \\ Q_{f,1} \\ Q_{f,2} \\ \boldsymbol{v} \end{bmatrix}^T = R \cdot \begin{bmatrix} Q_{i,0} \\ Q_{i,1} \\ Q_{i,2} \\ \boldsymbol{v} \end{bmatrix}^T \tag{44}$$

As discussed above, we need to guarantee the right-handedness property, thus, without loss generality, here we also assume the last two axes are reflected. The question turns to: does it exit another rotation matrix $R'$, such that:

$$\begin{bmatrix} Q_{f,0} \\ -Q_{f,1} \\ -Q_{f,2} \\ \boldsymbol{v} \end{bmatrix}^T = R' \cdot \begin{bmatrix} Q_{i,0} \\ Q_{i,1} \\ Q_{i,2} \\ \boldsymbol{v} \end{bmatrix}^T \tag{45}$$

We now use contradiction. Since we still have the two axes rotated 180 degrees around the first axes, $Q_{f,0}$, so $R' = (2Q_{f,0}Q_{f,0}^T - I)R$. Then given the two conditions $\boldsymbol{v}^T = R\boldsymbol{v}^T$ and $\boldsymbol{v}^T = R'\boldsymbol{v}^T$, we have $(2Q_{f,0}Q_{f,0}^T - I)\boldsymbol{v}^T = \boldsymbol{v}^T$.

If we let $Q_{f,0} = [k_1, k_2, k_3]$ and $\boldsymbol{v} = [v_1, v_2, v_3]$, then we have

$$(2Q_{f,0}Q_{f,0}^T - I)\boldsymbol{v}^T = \boldsymbol{v}^T$$

$$\begin{bmatrix} k_1 k_1 & k_1 k_2 & k_1 k_3 \\ k_1 k_2 & k_2 k_2 & k_2 k_3 \\ k_2 k_3 & k_2 k_3 & k_3 k_3 \end{bmatrix} \begin{bmatrix} v_1 \\ v_2 \\ v_3 \end{bmatrix} = \begin{bmatrix} v_1 \\ v_2 \\ v_3 \end{bmatrix}$$

$$\begin{bmatrix} k_1(k_1 v_1 + k_2 v_2 + k_3 v_3) \\ k_2(k_1 v_1 + k_2 v_2 + k_3 v_3) \\ k_3(k_1 v_1 + k_2 v_2 + k_3 v_3) \end{bmatrix} = \begin{bmatrix} v_1 \\ v_2 \\ v_3 \end{bmatrix} \qquad (46)$$

$$(k_1 v_1 + k_2 v_2 + k_3 v_3) \begin{bmatrix} k_1 \\ k_2 \\ k_3 \end{bmatrix} = \begin{bmatrix} v_1 \\ v_2 \\ v_3 \end{bmatrix}.$$

After calculation, we can obtain that $Q_{f,0} = c\boldsymbol{v}$, where $c$ is a coefficient. However, as we claimed in the condition, $\boldsymbol{v}$ does not lie in the same line as $Q_{f,0}$, thus, there does not exist such another rotation matrix $R' \neq R$ satisfying Equation (45). We also provide two examples in Figure 3 (c, d).

By contradiction, we can tell that there is only one unique rotation mapping from the initial inertial frame to the final inertial frame. □

To sum up, three points cannot form a rigid structure in Euclidean space, thus there can exist multiple reflection transformations, leading to opposite inertial frames. Four points can form a rigid structure, thus there exists only one reflection transformation.

# E    PROBLEM FORMULATION AND MORE DETAILS OF ASSEMBLEFLOW

## E.1    PROBLEM FORMULATION

We would like to emphasize that previous works aim at atomic level modeling, while our proposed AssembleFlow focuses on molecular level modeling. Meanwhile, both models need to satisfy the SE(3)-equivariance, as detailed below.

**Atomic Level Modeling**    Existing deep learning frameworks have been using atomic level modeling. For each atom $r$, the inference step is:

$$\boldsymbol{r}_{t+1} = \boldsymbol{r}_t + \boldsymbol{x}_{\theta,t},$$
$$\text{s.t. } \boldsymbol{x}_{\theta,t} \text{ is SE(3)-equivariant.} \tag{47}$$

Thus, we can observe that such a problem formulation cannot guarantee the rigidity of each molecule during the crystallization process.

**Molecular Level Modeling**    In our proposed AssembleFlow, it learns the translation and rotation at the molecule level. For each atom $r$, the inference step is:

$$\boldsymbol{r}_{t+1} = R_{\theta,t}(\boldsymbol{r}_t + \boldsymbol{x}_{\theta,t}),$$
$$\text{s.t. } \boldsymbol{x}_{\theta,t} \text{ and } R_{\theta,t} \text{ are SE(3)-equivariant.} \tag{48}$$

The $\boldsymbol{x}_{\theta,t}$ and $R_{\theta,t}$ are molecular level modeling. Notice that this also holds after we take the reparameterization, as discussed in Section 3.4. In Appendix E.2, we will explore how to define the SE(3)-equivariant models on top of that.

## E.2    SE(3)-EQUIVARIANT VELOCITY FUNCTION

We consider two types of SE(3)-equivariant models as the velocity function. As shown in Figure 7, the inputs are the same for learning: the positions at the initial step and the final step, respectively. We take the position of the mass center for each molecule in the cluster to obtain the translation in $\mathbb{R}^3$ ($\boldsymbol{x}_i$ and $\boldsymbol{x}_f$), and we take the first principal axes of inertial frames to obtain the reference coordinate system for rotation in SO(3) ($\boldsymbol{q}_i$ and $\boldsymbol{q}_f$ with alignment). Then we adopt Equations (6) and (7) for the interpolation on SO(3) and $\mathbb{R}^3$ group respectively, which gives us translation $\boldsymbol{x}_t$ and rotation $\boldsymbol{q}_t$ at interpolation time $t \in [0, 1]$.

Recall that the data structure considered here is the cluster of molecules, thus it is natural to split the modeling into intra-molecule and inter-molecule modeling, as introduced below.

**Intra-molecule Modeling.**    For each molecule in the cluster, we adopt the SE(3)-equivariant PaiNN (Schütt et al., 2021) to obtain the representation for each atom. Such an atomic representation can encode the inherent geometric structural information of individual molecules, which can be passed to inter-molecule modeling in the next step.

**Inter-molecule Modeling.**    This step aims to model the inter-molecule interactions during the molecular crystallization process based on the intra-molecule representation. We can have two options for SE(3)-equivariant inter-molecule modeling: (1) to project $\boldsymbol{x}_t$ and $\boldsymbol{q}_t$ back to obtain the atom-wise position and do modeling, as in Figure 7(a), or (2) to directly perform molecular level modeling on molecular-level translation $\boldsymbol{x}_t$ and rotation $\boldsymbol{q}_t$, as in Figure 7(b).

- **Atomic level modeling.** This means we build up the SE(3)-equivariant models on top of atom positions $\boldsymbol{r}_t$ at time $t$, and the outputs are the rotation and translation for step $t + 1$ or step $T$ (if we use reparameterization).
  - Obtain intra-molecule representation $\boldsymbol{h}_a$ using PaiNN (Schütt et al., 2021).
  - Obtain time embedding $\boldsymbol{h}_t$ with positional encoding (Vaswani et al., 2017).
  - Build up the vector frame basis (Liu et al., 2023) for each atom $F_a$, based on its neighborhoods.
  - Then we update the atomic representation $\boldsymbol{h}_a$ as the summation of $\boldsymbol{h}_a$ and $\boldsymbol{h}_t$.

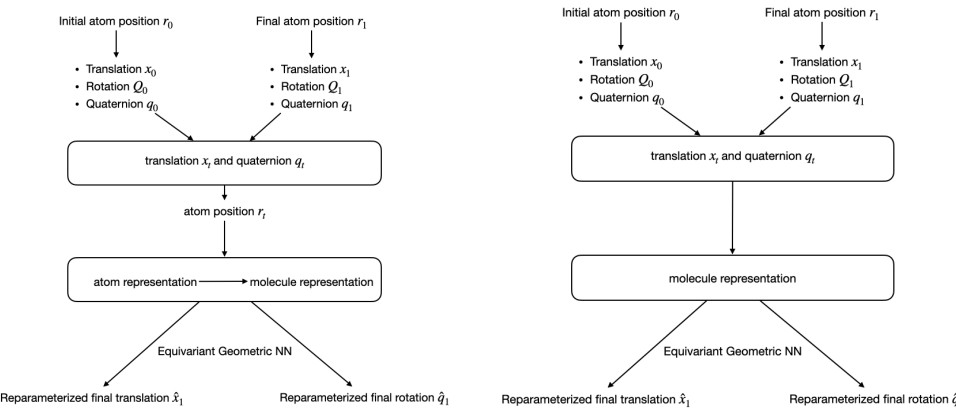

(a) Atomic level inter-molecule modeling.     (b) Molecular level inter-molecule modeling.

Figure 7: Illustration of two types of SE(3)-equivariant inter-molecule velocity functions.

– Apply message passing (Gilmer et al., 2017) to exchange the information between each atom and its neighborhood on top of the atomic representation $\boldsymbol{h}_a$ and vector frame basis $F_a$.

– The outputs include the atomic scalar representation $\boldsymbol{h}_{a,s} \in \mathbb{R}^{N \times d}$ and atomic vector representation $\boldsymbol{h}_{a,v} \in \mathbb{R}^{N \times 3 \times d}$, where $N$ is the total number of atoms in the cluster.

– Then we conduct the aggregation to obtain the molecular level predicted rotation velocity $\hat{\boldsymbol{q}}_\theta \in \mathbb{R}^{M \times 3}$ and predicted translation velocity $\hat{\boldsymbol{x}}_\theta \in \mathbb{R}^{M \times 3}$, where $M$ is the number of molecules in the cluster.

– This holds similarly if we are going to predict the final rotation $\hat{\boldsymbol{q}}_{1,\theta}$ and final translation vector $\hat{\boldsymbol{x}}_{1,\theta}$ when using the reparameterization.

• **Molecular level modeling.** This means we build up the SE(3)-equivariant models on top of $\boldsymbol{x}_t$ and $\boldsymbol{q}_t$ at time $t$, and the outputs are the rotation and translation for step $t + 1$ or step $T$ (if we are using reparameterization).

– Obtain intra-molecule representation $\boldsymbol{h}_a$ using PaiNN (Schütt et al., 2021).

– Obtain time embedding $\boldsymbol{h}_t$ with positional encoding (Vaswani et al., 2017).

– Then we update the atomic representation $\boldsymbol{h}_a$ as the summation of $\boldsymbol{h}_a$ and $\boldsymbol{h}_t$.

– Aggregate the atomic representation to get the molecular representation $\boldsymbol{h}_m$.

– Apply the message passing (Gilmer et al., 2017) to exchange the information between each molecule and other molecules in the cluster, where the interactions are treated as the forces in the inter-molecule level in the cluster.

– The outputs include a molecular level predicted rotation velocity $\hat{\boldsymbol{q}}_\theta \in \mathbb{R}^{M \times 3}$ and predicted translation velocity $\hat{\boldsymbol{x}}_\theta \in \mathbb{R}^{M \times 3}$, where $M$ is the number of molecules in the cluster.

– This holds similarly if we are going to predict the final rotation $\hat{\boldsymbol{q}}_{1,\theta}$ and final translation vector $\hat{\boldsymbol{x}}_{1,\theta}$ when using the reparameterization.

Notice that for the predicted quaternion $\hat{\boldsymbol{q}}_\theta$, we only predict the imaginary part. We then concat the real part as 1, followed by a normalization to make it a rotation quaternion.

### E.3 ALGORITHM

In this section, we provide the pseudocodes for AssembleFlow. Following Section 3, we illustrate the reparameterized version here. The training and inference algorithms are in Algorithms 1 and 2, respectively. We would also like to emphasize the following prior steps: (1) We first construct the inertial frames as discussed in the main article. (2) Then we construct the coordinate system (in rotation matrix) and conduct the alignment between the initial and final frames. (3) Last but not least, we transform the coordinate system from rotation matrix to quaternion as discussed in Appendix B.5.

---

**Algorithm 1** Learning of AssembleFlow.

---

1: **Inputs**: For $N$ atoms in $M$ molecules, we have atomic level initial position $r_0 \in \mathbb{R}^{N \times 3}$, molecular level initial rotation quaternion $q_0^{M \times 4}$ and translation $x_0 \in \mathbb{R}^{M \times 3}$, atomic level final position $r \in \mathbb{R}^{N \times 3}$, molecular level final rotation quaternion $q_1 \in \mathbb{R}^{M \times 4}$ and translation $x_1 \in \mathbb{R}^{M \times 3}$, timestep $T \in \mathbb{R}$, epoch $E \in \mathbb{R}$, coefficients $\alpha_0, \alpha_1 \in \mathbb{R}$.
2: **for** epoch $e \in [1, E]$ **do**
3:      Sample $t \in [1, T]$.
4:      Conduct LERP to obtain translation $x_t$ at time $t$, following Equation (7).
5:      Perform SLERP to obtain quaternion $q_t$ at time $t$, following Equation (6).
6:      Predict the final quaternion $\hat{q}_1 = \hat{q}_{1,\theta}(x_t, q_t, t)$ and translation $\hat{x}_1 = \hat{x}_{1,\theta}(x_t, q_t, t)$ using SE(3)-equivariant modeling as discussed in Appendix E.2.
7:      Minimize loss $\mathcal{L} = \alpha_0 \mathcal{L}_{\mathbb{R}^3,\text{reparameter}} + \alpha_1 \mathcal{L}_{\text{SO(3)},\text{reparameter}}$, as defined in Equation (13).
8: **end for**

---

---

**Algorithm 2** Inference of AssembleFlow.

---

1: **Inputs**: For $N$ atoms in $M$ molecules, we have atomic level initial position $r_0 \in \mathbb{R}^{N \times 3}$, molecular level initial rotation quaternion $q_0^{M \times 4}$ and translation $x_0 \in \mathbb{R}^{M \times 3}$, timestep $T \in \mathbb{R}$, learned SE(3)-equivariant models final rotation quaternion $\hat{q}_{1,\theta}(x_t, q_t, t)$ and translation $\hat{x}_{1,\theta}(x_t, q_t, t)$.
2: **for** timestep $t \in [1, T]$ **do**
3:      Predict the final quaternion $\hat{q}_1 = \hat{q}_{1,\theta}(x_t, q_t, t)$ and translation $\hat{x}_1 = \hat{x}_{1,\theta}(x_t, q_t, t)$ using SE(3)-equivariant modeling as discussed in Appendix E.2.
4:      Calculate the next-step quaternion $\hat{q}_{t+1}$ and translation $\hat{x}_{t+1}$ as Equation (14).
5:      Move the cluster of molecules w.r.t. $\hat{q}_{t+1}$ and $\hat{x}_{t+1}$.
6:      Obtain the corresponding atomic positions $r_{t+1}$ at time $t + 1$, following Equation (48).
7: **end for**
8: The final predicted crystal structure is $\hat{r}_T$.

---

### E.4 HYPER-PARAMETERS

We provide the key hyper-parameters of AssembleFlow in Table 6.

Table 6: Hyperparameter specifications for AssembleFlow.

| Model | Hyperparameter | Value |
|---|---|---|
| Intra-modeling | PaiNN | |
| | embedding dim | {128} |
| | num of layers | {3} |
| | cutoff | {5} |
| | read out | {mean} |
| Intra-modeling Atomic Level | num of layers | {2,5} |
| | num of convolution | {2} |
| | num of head | {4, 8} |
| | num of timesteps | {50, 200} |
| | $\alpha_0$ | {1} |
| | $\alpha_1$ | {1, 10} |
| Intra-modeling Molecular Level | num of layers | {4,5} |
| | num of head | {4, 8} |
| | num of timesteps | {50, 200} |
| | $\alpha_0$ | {1} |
| | $\alpha_1$ | {1, 10} |
| Optimization | seed | {0, 42, 123} |
| | epochs | {1000, 2000} |
| | cutoff $c$ | {20, 50} |
| | learning rate | {1e-4, 5e-4} |
| | optimizer | {Adam } |

## F    DETAILS OF BASELINES

### F.1    DEEP LEARNING BASELINES AND PARAMETERS

Notice that for all the baselines listed below, we also adopt the PaiNN for atomic level representation (Schütt et al., 2021), and the hyperparameters are the same as Appendix E. We list the remaining hyperparameters of baselines in Table 7.

Table 7: Hyperparameter specifications for deep learning baselines.

| Model | Hyperparameter | Value |
|---|---|---|
| GNN-MD | seed | {0, 42, 123} |
| | epochs | 3000 |
| | num of timesteps | {50, 200} |
| | cutoff $c$ | {20, 50} |
| | learning rate | {1e-4, 5e-4} |
| | optimizer | {Adam } |
| CrystalSDE | SDE type | {VE, VP} |
| | seed | {0, 42, 123} |
| | epochs | 3000 |
| | num of timesteps | {50, 200} |
| | cutoff $c$ | {20, 50} |
| | learning rate | {1e-4, 5e-4} |
| | optimizer | {Adam } |
| CrystalFlow | interpolation type | {VE, VP, LERP} |
| | seed | {0, 42, 123} |
| | epochs | 3000 |
| | num of timesteps | {50, 200} |
| | cutoff $c$ | {20, 50} |
| | learning rate | {1e-4, 5e-4} |
| | optimizer | {Adam } |

### F.2    PACKMOL

We want to mention that using PackMol  for evaluation with the packing matching metrics is nontrivial. This is because the atom ordering in each molecule and the molecule ordering in each cluster are different between PackMol simulated results and the ground-truth results. We note that deep learning methods do not have this issue because the orderings of initial atoms/molecules match with the final atoms/molecules.

Thus, to address this issue, we first use the Hungarian algorithm to match the mass centers of simulated results to obtain the least matching distance with the ground truth mass centers, *i.e.*, PM (center). This gives us the molecule ordering mapping from simulated clusters to the ground-truth clusters. Then for each molecule simulated-and-ground-truth pair, we apply the Hungarian algorithm again to obtain the minimum distance for alignment.

## G    DATASET AND EVALUATION METRICS

### G.1    DATASET

We evaluate our method using the crystallization dataset COD-Cluster17 from (Liu et al., 2024c). This COD-Cluster17 is a curated subset derived from the COD database (Grazulis et al., 2009). We note that some single molecules/substances can crystallize in different forms, known as polymorphs. This arises due to changes in the configurations during the process, such as the environment temperature, pressure, and solvent. COD-Cluster17 simplifies this setup by ignoring the configuration information and treats the crystallization problem as a density estimation problem.

### G.2    DETAILS OF EVALUATION METRICS

We illustrate five types of evaluation metrics below. Notice that in the original dataset, the dynamics or trajectories of molecules are missing. Thus, our evaluation is based on the ground truth cluster geometry at the last step.

**Packing Matching (PM)**    This metric quantifies how well the generated molecular assemblies match the reference crystal structures in terms of spatial arrangement and packing density (Chisholm & Motherwell, 2005). It is a key indicator of how accurately a model can replicate real-world crystallization patterns. We provide both the atomic MP, denoted as "PM (atom)" and mass-center-level PM, denoted as "PM (center)".

**Collision**    This follows (Cordero et al., 2008; Liu et al., 2024b). It measures if there is any atomic collision in the predicted assemblies. Atoms must maintain a minimum pairwise distance governed by the balance of attractive and repulsive forces. More concretely, we are using covalent radii as the most strict metric for atomic collisions in molecular generation. This is because it provides a precise lower bound for the distances between atoms when they are bonded. In other words, covalent radii represent the distance at which two atoms form a stable covalent bond, which is a very close and well-defined interaction compared to non-covalent interactions. However, other types of atomic radii, such as van der Waals radii or ionic radii, can be used for different purposes, depending on the nature of the interaction you're modeling.

**Separation**    We extend the metric from (Xie et al., 2022; Yang et al., 2024) to our setting. A cluster of molecules is valid if the minimum distance between molecules is above 0.5Å (Court et al., 2020). This metric is referred to as *separation* to measure the validity to avoid unphysical interactions at the molecular level.

**Compactness**    We propose this measure by calculating the percentage of simulated clusters where the maximum atomic pairwise distances are below 100Å. This assesses the spatial efficiency of the molecular assemblies, indicating how closely the constituent molecules are packed together. A higher compactness value suggests a more efficient arrangement, where the intermolecular spaces are minimized, leading to a denser crystalline structure.

# H  ABLATION STUDIES

## H.1  ABLATION STUDIES ON RANDOM SAMPLING

Here, we add another baseline by randomly sampling translation and rotation.

- For SO(3), we can do random sampling.
- For $\mathbb{R}^3$, we first obtain the range of atom positions in the training data, and then we just do uniform sampling within this range.

The results are in Table 8. As observed in Table 8, the Random baseline performs exceptionally well across all three validity metrics; however, its packing matching is significantly worse by an order of magnitude.

Table 8: AssembleFlow against six generative models on COD-Cluster17 with 5K, 10K, and all samples. The best results are marked in **bold**. Baseline Random has the best validity metrics, but they are meaningless since the packing matching is extremely high, remarking that the results collapse. Thus, we mark them in gray.

| | Packing Matching | | Validity | | |
|---|---|---|---|---|---|
| | PM (atom) ↓ | PM (center) ↓ | Collision ↓ | Separation ↑ | Compactness ↑ |
| Dataset: **COD-Cluster17-5K** | | | | | |
| Random | $54.07 \pm 0.42$ | $54.62 \pm 0.43$ | $0.31 \pm 0.01$ | $99.88 \pm 0.01$ | $100.00 \pm 0.00$ |
| GNN-MD | $13.67 \pm 0.06$ | $13.80 \pm 0.07$ | $27.53 \pm 0.49$ | $0.22 \pm 0.11$ | $\mathbf{100.00 \pm 0.00}$ |
| CrystalSDE-VE | $15.52 \pm 1.48$ | $16.46 \pm 0.99$ | $1.20 \pm 0.08$ | $27.17 \pm 0.86$ | $57.47 \pm 7.76$ |
| CrystalSDE-VP | $18.15 \pm 3.02$ | $19.15 \pm 4.46$ | $0.84 \pm 0.14$ | $53.13 \pm 12.89$ | $34.00 \pm 30.75$ |
| CrystalFlow-VE | $14.87 \pm 7.07$ | $13.08 \pm 4.51$ | $1.37 \pm 0.04$ | $35.70 \pm 0.73$ | $8.40 \pm 4.17$ |
| CrystalFlow-VP | $15.71 \pm 2.69$ | $17.10 \pm 1.89$ | $1.38 \pm 0.04$ | $35.43 \pm 0.88$ | $4.87 \pm 1.09$ |
| CrystalFlow-LERP | $13.59 \pm 0.09$ | $13.26 \pm 0.09$ | $0.34 \pm 0.01$ | $97.38 \pm 0.10$ | $\mathbf{100.00 \pm 0.00}$ |
| AssembleFlow (ours) | $\mathbf{7.27 \pm 0.04}$ | $\mathbf{6.13 \pm 0.10}$ | $\mathbf{0.33 \pm 0.00}$ | $\mathbf{97.64 \pm 0.36}$ | $\mathbf{100.00 \pm 0.00}$ |
| Dataset: **COD-Cluster17-10K** | | | | | |
| Random | $54.20 \pm 0.90$ | $54.76 \pm 0.90$ | $0.30 \pm 0.00$ | $99.86 \pm 0.01$ | $100.00 \pm 0.00$ |
| GNN-MD | $13.83 \pm 0.06$ | $13.90 \pm 0.05$ | $27.88 \pm 0.49$ | $0.23 \pm 0.11$ | $\mathbf{100.00 \pm 0.00}$ |
| CrystalSDE-VE | $17.25 \pm 2.46$ | $17.86 \pm 1.11$ | $0.99 \pm 0.27$ | $32.99 \pm 10.72$ | $34.93 \pm 14.99$ |
| CrystalSDE-VP | $22.20 \pm 3.29$ | $21.39 \pm 1.50$ | $0.53 \pm 0.35$ | $52.48 \pm 15.44$ | $16.83 \pm 18.09$ |
| CrystalFlow-VE | $16.41 \pm 2.64$ | $16.71 \pm 2.35$ | $1.42 \pm 0.03$ | $33.79 \pm 0.51$ | $5.47 \pm 0.47$ |
| CrystalFlow-VP | $19.39 \pm 4.37$ | $16.01 \pm 3.13$ | $1.44 \pm 0.03$ | $33.35 \pm 0.55$ | $4.23 \pm 0.48$ |
| CrystalFlow-LERP | $13.54 \pm 0.03$ | $13.20 \pm 0.03$ | $0.32 \pm 0.00$ | $97.32 \pm 0.05$ | $\mathbf{100.00 \pm 0.00}$ |
| AssembleFlow (ours) | $\mathbf{7.38 \pm 0.03}$ | $\mathbf{6.21 \pm 0.05}$ | $\mathbf{0.31 \pm 0.00}$ | $\mathbf{97.73 \pm 0.16}$ | $99.93 \pm 0.05$ |
| Dataset: **COD-Cluster17-All** | | | | | |
| Random | $65.94 \pm 0.07$ | $66.56 \pm 0.07$ | $0.30 \pm 0.00$ | $99.91 \pm 0.00$ | $100.00 \pm 0.00$ |
| GNN-MD | $22.30 \pm 12.04$ | $14.51 \pm 0.82$ | $24.29 \pm 4.58$ | $4.13 \pm 5.60$ | $98.77 \pm 1.73$ |
| CrystalSDE-VE | $17.28 \pm 0.73$ | $18.92 \pm 0.03$ | $0.19 \pm 0.18$ | $15.47 \pm 12.42$ | $2.51 \pm 2.37$ |
| CrystalSDE-VP | $18.03 \pm 4.56$ | $20.02 \pm 3.70$ | $0.55 \pm 0.19$ | $48.78 \pm 1.70$ | $6.88 \pm 2.82$ |
| CrystalFlow-VE | $12.80 \pm 1.20$ | $15.09 \pm 0.34$ | $1.41 \pm 0.01$ | $35.34 \pm 0.28$ | $2.90 \pm 0.02$ |
| CrystalFlow-VP | $13.50 \pm 0.44$ | $13.28 \pm 0.48$ | $1.51 \pm 0.02$ | $33.06 \pm 1.31$ | $6.61 \pm 3.17$ |
| CrystalFlow-LERP | $13.61 \pm 0.00$ | $13.28 \pm 0.01$ | $0.34 \pm 0.00$ | $97.34 \pm 0.02$ | $\mathbf{99.99 \pm 0.01}$ |
| AssembleFlow (ours) | $\mathbf{7.37 \pm 0.01}$ | $\mathbf{6.21 \pm 0.01}$ | $\mathbf{0.31 \pm 0.00}$ | $\mathbf{98.15 \pm 0.22}$ | $99.98 \pm 0.00$ |

## H.2 ABLATION STUDIES ON INTERPOLATION ON SO(3)

**Empirical results.** We consider replacing the SLERP with EMLERP in AssembleFlow, and name it as AssembleFlow-EMLERP. We conduct the experiment on COD-5000, where we are taking the optimal hyperparameters from AssembleFlow.

The results are in Table 9. As observed, using SLERP is better than EMLERP. We are still running results for COD-10k and COD, and will update the results later.

Table 9: AssembleFlow against six generative models on COD-Cluster17 with 5K, 10K, and all samples. The best results are marked in **bold**.

| | Packing Matching | | Validity | | |
|---|---|---|---|---|---|
| | PM (atom) ↓ | PM (center) ↓ | Collision ↓ | Separation ↑ | Compactness ↑ |
| Dataset: **COD-Cluster17-5K** | | | | | |
| GNN-MD | $13.67 \pm 0.06$ | $13.80 \pm 0.07$ | $27.53 \pm 0.49$ | $0.22 \pm 0.11$ | $\mathbf{100.00 \pm 0.00}$ |
| CrystalSDE-VE | $15.52 \pm 1.48$ | $16.46 \pm 0.99$ | $1.20 \pm 0.08$ | $27.17 \pm 0.86$ | $57.47 \pm 7.76$ |
| CrystalSDE-VP | $18.15 \pm 3.02$ | $19.15 \pm 4.46$ | $0.84 \pm 0.14$ | $53.13 \pm 12.89$ | $34.00 \pm 30.75$ |
| CrystalFlow-VE | $14.87 \pm 7.07$ | $13.08 \pm 4.51$ | $1.37 \pm 0.04$ | $35.70 \pm 0.73$ | $8.40 \pm 4.17$ |
| CrystalFlow-VP | $15.71 \pm 2.69$ | $17.10 \pm 1.89$ | $1.38 \pm 0.04$ | $35.43 \pm 0.88$ | $4.87 \pm 1.09$ |
| CrystalFlow-LERP | $13.59 \pm 0.09$ | $13.26 \pm 0.09$ | $0.34 \pm 0.01$ | $97.38 \pm 0.10$ | $\mathbf{100.00 \pm 0.00}$ |
| CrystalFlow-LERP | $13.59 \pm 0.09$ | $13.26 \pm 0.09$ | $0.34 \pm 0.01$ | $97.38 \pm 0.10$ | $\mathbf{100.00 \pm 0.00}$ |
| AssembleFlow-EMLERP | $7.30 \pm 0.04$ | $6.32 \pm 0.04$ | $0.37 \pm 0.01$ | $93.38 \pm 0.54$ | $\mathbf{100.00 \pm 0.00}$ |
| AssembleFlow (ours) | $\mathbf{7.27 \pm 0.04}$ | $\mathbf{6.13 \pm 0.10}$ | $\mathbf{0.33 \pm 0.00}$ | $\mathbf{97.64 \pm 0.36}$ | $\mathbf{100.00 \pm 0.00}$ |
| Dataset: **COD-Cluster17-10K** | | | | | |
| GNN-MD | $13.83 \pm 0.06$ | $13.90 \pm 0.05$ | $27.88 \pm 0.49$ | $0.23 \pm 0.11$ | $\mathbf{100.00 \pm 0.00}$ |
| CrystalSDE-VE | $17.25 \pm 2.46$ | $17.86 \pm 1.11$ | $0.99 \pm 0.27$ | $32.99 \pm 10.72$ | $34.93 \pm 14.99$ |
| CrystalSDE-VP | $22.20 \pm 3.29$ | $21.39 \pm 1.50$ | $0.53 \pm 0.35$ | $52.48 \pm 15.44$ | $16.83 \pm 18.09$ |
| CrystalFlow-VE | $16.41 \pm 2.64$ | $16.71 \pm 2.35$ | $1.42 \pm 0.03$ | $33.79 \pm 0.51$ | $5.47 \pm 0.47$ |
| CrystalFlow-VP | $19.39 \pm 4.37$ | $16.01 \pm 3.13$ | $1.44 \pm 0.03$ | $33.35 \pm 0.55$ | $4.23 \pm 0.48$ |
| CrystalFlow-LERP | $13.54 \pm 0.03$ | $13.20 \pm 0.03$ | $0.32 \pm 0.00$ | $97.32 \pm 0.05$ | $\mathbf{100.00 \pm 0.00}$ |
| AssembleFlow-EMLERP | $7.51 \pm 0.17$ | $6.46 \pm 0.22$ | $0.33 \pm 0.00$ | $94.68 \pm 0.44$ | $99.93 \pm 0.05$ |
| AssembleFlow (ours) | $\mathbf{7.38 \pm 0.03}$ | $\mathbf{6.21 \pm 0.05}$ | $\mathbf{0.31 \pm 0.00}$ | $\mathbf{97.73 \pm 0.16}$ | $99.93 \pm 0.05$ |
| Dataset: **COD-Cluster17-All** | | | | | |
| GNN-MD | $22.30 \pm 12.04$ | $14.51 \pm 0.82$ | $24.29 \pm 4.58$ | $4.13 \pm 5.60$ | $98.77 \pm 1.73$ |
| CrystalSDE-VE | $17.28 \pm 0.73$ | $18.92 \pm 0.03$ | $0.19 \pm 0.18$ | $15.47 \pm 12.42$ | $2.51 \pm 2.37$ |
| CrystalSDE-VP | $18.03 \pm 4.56$ | $20.02 \pm 3.70$ | $0.55 \pm 0.19$ | $48.78 \pm 1.70$ | $6.88 \pm 2.82$ |
| CrystalFlow-VE | $12.80 \pm 1.20$ | $15.09 \pm 0.34$ | $1.41 \pm 0.01$ | $35.34 \pm 0.28$ | $2.90 \pm 0.02$ |
| CrystalFlow-VP | $13.50 \pm 0.44$ | $13.28 \pm 0.48$ | $1.51 \pm 0.02$ | $33.06 \pm 1.31$ | $6.61 \pm 3.17$ |
| CrystalFlow-LERP | $13.61 \pm 0.00$ | $13.28 \pm 0.01$ | $0.34 \pm 0.00$ | $97.34 \pm 0.02$ | $\mathbf{99.99 \pm 0.01}$ |
| AssembleFlow-EMLERP | $\mathbf{7.28 \pm 0.00}$ | $6.23 \pm 0.01$ | $0.35 \pm 0.00$ | $93.17 \pm 0.02$ | $99.98 \pm 0.00$ |
| AssembleFlow (ours) | $7.37 \pm 0.01$ | $\mathbf{6.21 \pm 0.01}$ | $\mathbf{0.31 \pm 0.00}$ | $\mathbf{98.15 \pm 0.22}$ | $99.98 \pm 0.00$ |

