# OpenReview forum: "AssembleFlow: Rigid Flow Matching with Inertial Frames for Molecular Assembly"
_ICLR.cc/2025/Conference — ICLR 2025 Poster_

### Official Review · Reviewer_cXch · 2024-10-27

**Soundness:** 2
**Presentation:** 2
**Contribution:** 2
**Rating:** 6
**Confidence:** 4

**Summary:**

This paper proposes AssembleFlow, which models the assembly of molecular crystals by assuming a rigid body and operating in the SE(3) space. It learns vector fields on SO(3) and $\mathbb{R}^3$ separately, using SLERP and LERP, respectively. The reference coordinates systems are constructed with inertial frames.

**Strengths:**

- The paper is written clearly and is easy to understand.
- The experiment shows good performance.

**Weaknesses:**

A notable limitation of this work is the claim to be the first to implement rigid generation in SE(3) space for molecular assembly. This claim appears to overlook prior work [1], which closely aligns with the approach presented here, treating molecules as rigid bodies in molecular assembly, separately handling SO(3) and Euclidean space, and applying quaternion representation on SO(3). The differences are minimal, mainly in the methods for constructing rigid frames and in the choice of a normalizing flow over a CNF with flow matching.

Additionally, the paper emphasizes the decomposition of SE(3) into $\mathbb{R}^3$ and SO(3) as a unique strength (e.g., “this decomposition also empowers distinct probability paths…, effectively allowing for separate learning”). However, decomposing SE(3) into SO(3) and $\mathbb{R}^3$ is a widely adopted simplification in SE(3)-based methods ([1,2] among others). To my knowledge, there are no approaches that directly operate on SE(3) without this decomposition, which raises questions about why this common approach is presented as an innovative strength.

Given these points, I am hesitant to support this work for acceptance in its current form.

-----
[1] Rigid Body Flows for Sampling Molecular Crystal Structures, Köhler et al., ICML 2023
[2] SE(3)-Stochastic Flow Matching for Protein Backbone Generation, Bose et al., ICLR 2024

**Questions:**

What advantage does the eigendecomposition of the inertial tensor $\hat{I}$ have over the common approach of using centered $X$?

---

> ### Author Response · Authors · 2024-11-20
> **Reply to Reviewer cXch (part 1/2)**
>
> Thank you for your thorough review of our paper. We have carefully addressed the concerns you raised, and you can find the corresponding responses below.
>
>
> First for notation, let us use $X_1$ and $X_0$ be the strongly correlated structures (stable) and weakly correlated structures (unstable), respectively.
>
>
> **Q: A notable limitation of this work is the claim to be the first to implement rigid generation in SE(3) space for molecular assembly.**
>
> We believe this claim is accurate, as our work is the first to implement SE(3)-based rigid generation specifically for molecular assembly, focusing on modeling dynamic processes rather than stable state distributions. While the ICML work [1] provides valuable contributions to equilibrium conformational sampling, our approach addresses a complementary aspect by capturing the dynamic processes and distributions associated with molecular assembly. We see these efforts as addressing distinct but equally important challenges in molecular modeling.
>
>
> **Q: The differences between our work and the ICML 2023 paper [1]**
>
> Thank you for pointing out this paper [1], we agree that this is relevant, and have added a section (Sec A.5) to discuss the difference between the paper and our work in the revised version. Meanwhile, we would like to emphasize that our work differs from the ICML 2023 paper [1] in three fundamental aspects:
>
> 1. **Dynamic Transition Modeling vs. Stable Structure Sampling:** Unlike [1] which focuses on stable structure sampling, our work models the dynamic transition process from weakly correlated (unstable) structures to strongly correlated (stable) structures. Namely, the objectives are different: [1] models $p(X_1)$ while AssembleFlow models $p(X_1|X_0)$ (Fig 1). Notably, dynamic transition modeling toward stability is identified as **a nontrivial next step in the ICML work [1]**, which states 'An interesting but
> nontrivial next step would be ... a flow model for the positions that can handle ... phase transitions'.
> 2. **Use of Inertial Frames for Rigid Modeling:** The ICML paper [1] '**assumes** that these bodies are rigid'. In contrast, we employ inertial frames to **enable rigid transformations**, including both translation and rotation. This approach is also suggested as **a future direction in the ICML paper**, which states 'exploiting the SE(3) symmetry of jointly moving all rigid bodies'. Achieving SE(3) equivariance for rigid transformations in materials is **non-trivial**, as it **handling permutation, and applying a reparameterization** technique to directly model SE(3) at time t.
> 3. **Limitations of Normalizing Flow Models on Molecular Assembling:** ~~Normalizing flows require a Gaussian prior, which is unsuitable for our weakly correlated structures that typically do not follow a normal distribution. The ICML paper also faces challenges with interpolation in dynamic modeling, as normalizing flows do not inherently support interpolation.~~ Normalizing flows require a tractable prior (like Gaussian), which is non-trivial for weakly correlated or unstable structures. The ICML paper also faces a similar challenge and it solves it by introducing a surrogate prior for sampling equilibrium molecules (not unstable) with MD simulation. On the contrary, our proposed AssembleFlow is built on the Flow Matching framework, which supports the interpolation for dynamics simulation. In AssembleFlow, the LERP and SLERP interpolation methods enable us to model such rigid transformations between two arbitrary states, including from an unstable state to a stable state.
>
> [1] Rigid Body Flows for Sampling Molecular Crystal Structures, Köhler et al., ICML 2023

---

> ### Author Response · Authors · 2024-11-20
> **Reply to Reviewer cXch (part 2/2)**
>
> **Q: Related work [2].**
>
> Thank you for pointing out this paper by Bose [2]. We have already discussed this work (and relevant works) in the Sec A.3 in the first version and a simlar answer to reviewer 1PjM. We agree that decomposing SE(3) into SO(3) and R(3) is not novel, yet we would like to emphasize that such a decomposition on inertial frame and with LERP and SLERP is novel. We summarize the main differences below, and you can find more details in Sec A.4 in the revised version.
>
> | Feature| AssembleFlow | FoldFlow |
> |----|----|----|
> | Reference Frame | Eigenvectors of inertial frames | Gram-Schmidt on $N$-$C$-$C_{\alpha}$|
> | SO(3) Interpolation| Spherical Interpolation (SLERP) | Exponential Map Interpolation (MELERP) |
> | Equation | $\frac{\sin((1-t)\omega) q_0 + \sin(t \omega) q_1}{\sin(\omega)}$ | $\exp_{r_0} (t \log_{r_0} (r_1))$ |
> | Derivative | Eq. 9 in paper| Eq. 16 in paper|
> | Rotation Representation (for Velocity / Objective Function) | Quaternion| Rotation Matrix or Axis-angle |
> | Reparameterization | Yes| No |
>
> Besides, we further conduct an ablation study on COD-5000 below, and as observed, using SLERP is better than MELERP. Please check Sec H.2 for more details. BTW. We are still running experiments on COD-10000 and COD, and will update the results soon.
>
> | |PM(atom)|PM(center)|Collision|Separation|Compactness|
> |----|----|----|----|----|----|
> |AssembleFlow-MELERP|7.30±0.04|6.32±0.04|0.37±0.01|93.38±0.54|100.00±0.00|
> |AssembleFlow (ours)|7.27±0.04|6.13±0.10|0.33±0.00|97.64±0.36|100.00±0.00|
>
> [2] SE(3)-Stochastic Flow Matching for Protein Backbone Generation, Bose et al., ICLR 2024
>
> **Q: What advantage does the eigendecomposition of the inertial tensor I have over the common approach of using centered X.**
>
> Thank you for raising this question. This is indeed one of the biggest contributions of our work: utilizing inertial frame for rigid modeling. **Using $I$ enables rigid modeling in AssembleFlow** while **centered $X$ in baseline CrystalFlow cannot**:
> - Using centered $X$ cannot guarantee rigidity. This is because we are modeling the atom-level rotation from $X_0$ to $X_1$, which cannot guarantee the rigidity. The baselines CrystalFlow and its variants are simulating crystallization directly in this way, without using inertial frames, and you can see the performance gap between them.
> - Using $I$ can guarantee rigidity. This is because instead of modeling all the atoms, AssembleFlow learns how the reference frame rotates from $X_0$ to $X_1$.
> - To sum up, the key difference is that **learning rotation on the refernece frame can guarantee rigidity**.
>
> BTW. Can you help list the papers that are using centered $X$ for handling the molecular crystallization or rigid modeling? We are happy to check in more details on these any other related works.

---

> ### Comment · Reviewer_cXch · 2024-11-24
>
> Thank you for your response. My key concerns are still unresolved. Namely,
>
> **1. Differences between [1] and the proposed work**
>
> The authors state:
>
> > Our work models the dynamic transition process from weakly correlated (unstable) structures to strongly correlated (stable) structures
>
> However, the provided mathematical formulation $p(X_1|X_0)$ does not model the dynamic transition process. The term 'dynamic transition' typically refers to modeling the entire trajectory from $X_0$ to $X_1$, as in molecular dynamics [2, 3]. This work, however, simply adds a conditioning term $X_0$ to $p(X_1)$ without modeling the intermediate transition process.
>
> Additionally, I have reviewed the construction of the COD-Cluster17 dataset and observed that $X_0$ is generated by randomly applying rototranslations to $X_1$, rather than simulating a trajectory (using molecular dynamics) from a randomly initialized $X_0$ to a stable state $X_1$. This raises concerns about whether this dataset is suitable for addressing the task of molecular assembly, as $X_0$ and $X_1$ may not accurately represent a true pair in the assembly process. If I have misunderstood, I kindly ask for clarification.
>
> > Normalizing flows require a Gaussian prior, which is unsuitable for our weakly correlated structures that typically do not follow a normal distribution.
>
> This claim is inaccurate. Normalizing flows do not require a Gaussian prior, as evidenced in the cited paper [1]. While I recognize the advantages of flow matching over traditional normalizing flows, this statement raises concerns about whether the authors have fully understood the difference between their approach and [1].
>
> Additionally, I noticed the absence of a reproducibility statement and a link to the code. Including these would be greatly helpful for better understanding the differences and ensuring the work's reproducibility.
>
> **2. Presentation of the paper.**
>
> > Thank you for pointing out this paper by Bose [2]. We have already discussed this work (and relevant works) in the Sec A.3 in the first version and a simlar answer to reviewer 1PjM. We agree that decomposing SE(3) into SO(3) and R(3) is not novel, yet we would like to emphasize that such a decomposition on inertial frame and with LERP and SLERP is novel.
>
> I am aware of the difference between this work and that of Bose. My concern lies with the presentation of the paper. While the authors acknowledge that the decomposition is not novel, statements such as "This decomposition also empowers..." and "Such decomposition allows the explicit enforcement of ..." give the impression that it is a novel approach and is their contribution. I believe the authors could improve the presentation to more clearly distinguish their contributions from existing work.
>
> **3. Advantage inertial frames over PCA.**
>
> > Thank you for raising this question. This is indeed one of the biggest contributions of our work: utilizing inertial frame for rigid modeling.
>
> Regarding my statement on centered $X$, I was asking about the advantage of inertial frames over common approaches of using PCA. I apologize for the ambiguity in my earlier question.
>
> While I appreciate the authors' response to Reviewer 1PjM regarding the numerical instability of PCA, the experiment setup seems somewhat artificial and unclear (e.g., "Notice that since the MSE reconstruction is meaningless when it is too small (why?), so we only compare these two frames when at least one of them has reconstruction greater than or equal to a threshold θ."). Given the authors' claim that this is the biggest contribution, the significance of inertial frames would be more evident if they provided ablations comparing the performance using PCA versus inertial frames.
>
> ---
> [1] Rigid Body Flows for Sampling Molecular Crystal Structures, Köhler et al., ICML 2023.
> [2] Timewarp: Transferable Acceleration of Molecular Dynamics by Learning Time-Coarsened Dynamics, Klein et al., NeurIPS 2023.
> [3] Generative Modeling of Molecular Dynamics Trajectories, Jing et al., NeurIPS 2024.

---

> ### Author Response · Authors · 2024-11-25
> **Reply**
>
> | | Paper [1] | AssembeFlow |
> | --- | --- | --- |
> | Experiments | Transition between two stable conformations | Transition from unstable to stable |
> | Data | Water and methane (experiment missing on GitHub repo) molecules | COD organic molecules |
> | Frame construction | H-O-H frame specific for water molecule (No code or comments related to methane rigid modeling were found in the GitHub repo) | Inertial frame for any organic molecule |
> | Objective Function | Upper bound of log-ratio for energy difference estimation | Direct estimation of conditional density from unstable to stable conformation |
> | Modeling SE(3) symmetry in moving rigid bodies | No | Yes |
> | Avoid bijectivity constraint in coupling layer | No | Yes |
>
>
> **Q. Differences between [1] and the proposed work**
>
> Thank you for the feedback. We were initially confused by [1], as it blends the preliminaries (Eqs. 2, 4, and 5) with the core objectives (Eqs. 3 and 7). By Gaussian, we meant the tractable priors in general. [1] discussed using Gaussian as priors but replaced it with samples drawn from equilibrium MD simulation.
>
> Now after revisiting [1] for further details, we have fixed our statement in previous round and provided our updated comparison below. Additionally, we would like to emphasize that our work can be seen as an extension of [1], yet addressing more practical and challenging problems, including rigid modeling on arbitrary molecules, SE(3)-equivariant modeling, and interpolation modeling.
> - [1] targets at modeling the transition, e.g., water molecules at different temperatures. The MD simulation can provide samples at the stable or equilibrium status. Our work models the transition from the unstable to stable state.
> - [1] specifically models the frame for H-O-H [(codes here)](https://github.com/noegroup/rigid-flows/blob/main/rigid_flows/rigid.py#L13-L29) (no code or comments related to methane rigid modeling were found in the GitHub repository, only [ice](https://github.com/noegroup/rigid-flows/tree/main/experiments)). In contrast, our approach is more generalizable, as the inertial frame can serve as a reference frame for any molecule. Thus, [1] cannot be directly applied to our dataset, but our work can be viewed as an extension of [1] to a more general setting.
> - [1] estimates an upper bound of log-ratio for energy difference estimation between two equilibrium states, while ours is modeling density estimation directly for unstable to stable transition.
> - [1] states it does not 'explore the SE(3) symmetry of moving all rigid bodies', and we solve this limitation by introducing the SE(3)-equivariant modeling from two granularities.
> - [1] does modeling with an extra bijectivity constraint, limiting the model capacity (e.g., Flow++). The velocity function models in FM do not have this issue.
> The transition of a molecular cluster from weakly correlated (unstable) structures to strongly correlated (stable) structures is a special case of the general dynamics. The limitation here is the data insufficiency (lack of intermediate snapshots), not modeling.
>
> **Q. Presentation of the paper**
>
> We will improve our presentation as suggested.
>
> **Q. Advantage inertial frames over PCA**
>
> We have proven that using the inertial frame is more accurate than using PCA. This is the reconstruction experiment in reply to Reviewer 1PjM. Please feel free to check the details.
>
> When $\delta_{\text{Inertial}}$ or $\delta_{\text{PCA}}$ gets too small, as on the order of 1e-12, it falls well below the sensitivity of the dataset, rendering the comparison meaningless. Therefore, we consider the threshold from 1e-3 to 1e-8 for clearer and more meaningful comparison.

---

> > ### Comment · Reviewer_cXch · 2024-11-27
> >
> > Thank you for your response. My concerns have been partially addressed, and I have raised my score to 5. However, the score remains limited because some of my earlier comments were not addressed. Could the authors share their thoughts on the following points:
> > - Why do the authors consider this dataset suitable for molecular assembly, given that a pair might not reflect the true assembly process?
> > > Additionally, I have reviewed the construction of the COD-Cluster17 dataset and observed that $X_0$ is generated by randomly applying rototranslations to $X_1$, rather than simulating a trajectory (using molecular dynamics) from a randomly initialized $X_0$ to a stable state $X_1$. This raises concerns about whether this dataset is suitable for addressing the task of molecular assembly, as $X_0$ and $X_1$ may not accurately represent a true pair in the assembly process. If I have misunderstood, I kindly ask for clarification.
> > - Will the authors provide the code?
> > > Additionally, I noticed the absence of a reproducibility statement and a link to the code. Including these would be greatly helpful for better understanding the differences and ensuring the work's reproducibility
> > - Why did the authors choose this experimental setting, instead of performing ablations on the main experiments for performance comparison?
> > > While I appreciate the authors' response to Reviewer 1PjM regarding the numerical instability of PCA, the experiment setup seems somewhat artificial and unclear (e.g., "Notice that since the MSE reconstruction is meaningless when it is too small (why?), so we only compare these two frames when at least one of them has reconstruction greater than or equal to a threshold θ."). Given the authors' claim that this is the biggest contribution, the significance of inertial frames would be more evident if they provided ablations comparing the performance using PCA versus inertial frames.

---

> ### Author Response · Authors · 2024-11-28
> **Reply**
>
> Thank you! We are happy to share additional insights.
>
> **Q: Why COD-Cluster17.**
>
> Thank you for raising this question. We actually asked a similar question to the authors of the COD-Cluster17, and one of the authors, a crystallographer, confirmed that this geometrically optimized dataset closely aligns with what crystallographers would typically use in practice, for their wet-lab experiments.
>
> Additionally, we would like to share a critical comment from authors of COD-Cluster17, which is related to your question, is: "What constitutes `true packing dynamcis`? Because we think the only true dynamics is what is happenging in reallity, and we are only able to capture that by a super-powerful electron microscope, which is not yet practical." They also stated, "From this aspect, we do not consider MD simulation to represent true dynamcis." This is the general viewpoint shared by them (especially from a wet-lab crystallographer), and we believe it holds some merit. Feel free to reach out to the authors for further insights if needed.
>
> On the other hand, our algorithm is designed to generalize to other datasets with diverse molecular packing dynamics. An ideal dataset would include more snapshots along packing trajectories and encompass a wider variety of molecular systems. We hope our work can inspire more people into exploring this problem and contribute to the development of additional datasets.
>
> BTW. If you have any recommendations for alternative datasets, we would greatly appreciate your input.
>
> **Q: Code release.**
>
> Yes, we will release the code to support open science.
>
> **Q: Why experiment on stability of frame construction.**
>
> The experiment we designed and conducted aims to measure the stability of using an inertial frame and PCA to construct the reference frame, representing the most straightforward approach for comparison. While we could have conducted complete experiments based solely on PCA, we didn't as the stability comparison is sufficient to demonstrate that the inertial frame is more stable than PCA.
>
> Now that we have a few more days during the rebuttal period, we could try our best to add the complete experiment if you insist. However, before proceeding, we would like to double-check whether you feel the stability comparison alone is insufficient.

---

> > ### Comment · Reviewer_cXch · 2024-11-28
> >
> > I find the dataset author's statement, 'From this aspect, we do not consider MD simulation to represent true dynamics,' quite concerning, as it undermines the entire field of molecular dynamics and even part of theoretical physics.
> >
> > That said, I agree with the author's claim that their method can work well on other datasets. I am fine with more experiments and have raised my score to 6.

---

> > > ### Author Response · Authors · 2024-12-02
> > > **Reply**
> > >
> > > Thank you for your comments. The crystallographer's comments were specifically referring to the molecular crystallization problem. These insights may not necessarily apply to other MD simulation applications. We hope this clarification helps alleviate your concerns to some extent.
> > >
> > > We have finished the new experiments of COD-5k, using two frames. The results are attached below:
> > > | |PM(atom)|PM(center)|Collision|Separation|Compactness|
> > > |----|----|----|----|----|----|
> > > |AssembleFlow-Inertial Frame|7.27±0.04|6.13±0.10|0.33±0.00|97.64±0.36|100.00±0.00|
> > > |AssembleFlow-PCA | 7.29±0.07 | 6.22±0.07 | 0.35±0.00 | 95.98±1.06 | 100.00±0.00|

---

### Official Review · Reviewer_fnyR · 2024-11-03

**Soundness:** 4
**Presentation:** 4
**Contribution:** 3
**Rating:** 6
**Confidence:** 5

**Summary:**

The paper introduces AssemblyFlow, a flow-matching based generative model to predict the packing of molecules into crystalline form. The method decouples the prediction of crystallization into 2 orthogonal bases, the rotation and the translation of molecular frames. A linear and spherical interpolation scheme are derived for each base, respectively. The score model is parameterized using a SE(3) equivariant network. The model is benchmarked against both generative model and simulation model baselines. Benchmarks and evaluation metrics are developed to evaluate the model performance. AssembleFlow achieves better performance compared with generative model baselines. Compared with simulation baseline PackMol, it achieves comparable performance but is 25 fold faster.

**Strengths:**

- The paper introduces a rigid frame based flow matching model to generate the structure of organic crystals. The approach is similar to those introduced for frame-based flow matching methods for protein backbone generative, like FrameFlow [1]. Unlike protein, the frame for organic molecules is not well defined. The novelty of this approach is to use the inertial frames of the reference frames for organic molecules.
- A spherical interpolation scheme is introduced to model the rotation of molecular frames. A quaternion formalism is used to define the interpolation in the SO(3) group. The formalism looks similar to those used for protein frame modeling. It is unclear to me what is the core difference in the author’s approach.
- A set of new metrics, including packing matching (PM), atomic collision, separation, and compactness are introduced for the crystallization packing problem. The set of metrics are novel and meaningful from the domain perspective. It will benefit the community to further develop methods in the field.
- The signficant improvements of AssembleFlow compared with other generative model baselines show the advantage of modeling the problem as rigid frames. This advantage is one of the core innovations of the method.
- AssembleFlow is able to match the performance of MolPack, a simulation baseline. This is quite significant. The 25 fold speed-up compared with MolPack shows the advantage of generative modeling approach.

[1] Yim, Jason, et al. "Fast protein backbone generation with SE (3) flow matching." arXiv preprint arXiv:2310.05297 (2023).

**Weaknesses:**

- The decoupling of SE(3) into R^3 and SO(3) groups, i.e., rigid flow matching, has been extensively explored in the protein backbone community. The authors need to review more related papers and articulate their innovations in the methodology.
- As the authors pointed out, numerical stability exists when there is a tie in eigenvalues for the principal axes of inertia. This problem is especially important for symmetric molecules. Can the authors add more details regarding how they address the instability issue? How much will it influence the training stability?
- The real organic crystals are periodic. However, the authors only model them as non-periodic clusters in this work. It is a foundational limitation of both the dataset and the AssembleFlow model.
- In real organic crystals, the rigid assumption may not always apply. The molecular conformation may change. Therefore, the rigid frame modeling approach may be limited in predicting the structure of more flexible molecules.

**Questions:**

- Can the authors explain the difference between their quaternion interpolation and the other SE(3) interpolation scheme like those developed in FrameFlow? Do you expect a difference in performance if you adopt a different SE(3) interpolation scheme?
- In Fig. 4, it is very clear that PackMol generated structures have a much lower energy compared with AssembleFlow. This indicates a limitation of the generated structures. Can the authors expand on why AssembleFlow generated structures have a much higher energy?
- The authors mention 2 versions of their SE(3)-equivariant network for inter-molecule modeling. I could not find more results on comparing these 2 versions. Can the authors add more results or point me to the right location?
- How did the authors predict the initial conformation of the individual molecules? Are those predicted by RDKit? Or do you directly take the conformation from the data?

---

> ### Author Response · Authors · 2024-11-20
> **Reply to Reviewer fnyR (part 1/2)**
>
> Thank you for recognizing the novelty of using the inertial frame as a reference, the meaningful metrics, the significant improvements over deep learning baselines, and the efficiency compared to numerical methods. We also appreciate your detailed questions, particularly those regarding the protein rigid modeling comparison. We have provided the responses below.
>
> **Q: Comparison of two interpolation methods.**
>
> Thank you for raising this critical question (we have a similar answer to reviewer 1PjM). These relevant papers (FoldFlow and FrameFlow) are using **exponential map interpolation** (EMLERP) for SO(3) interpolation.
>
> SLERP and EMLERP are different interpolation methods on SO(3), and they do not have a clear methodological advantage over each other; however, the EMLERP considers more approximation tricks in implementation (FoldFlow and FrameFlow). We provide a detailed comparison below, and you can find more details in Sec A.4 in the revised version.
>
> | Feature| AssembleFlow | FoldFlow |
> |----|----|----|
> | Reference Frame | Eigenvectors of inertial frames | Gram-Schmidt on $N$-$C$-$C_{\alpha}$|
> | SO(3) Interpolation| Spherical Interpolation | Exponential Map Interpolation |
> | Equation | $\frac{\sin((1-t)\omega) q_0 + \sin(t \omega) q_1}{\sin(\omega)}$ | $\exp_{r_0} (t \log_{r_0} (r_1))$ |
> | Derivative | Eq. 9 in paper| Eq. 16 in paper|
> | Rotation Representation (for Velocity / Objective Function) | Quaternion| Rotation Matrix or Axis-angle |
> | Reparameterization | Yes| No |
>
> Besides, we further conduct an ablation study on COD-5000 below, and as observed, using SLERP is better than MELERP. Please check Sec H.2 for more details. BTW. We are still running experiments on COD-10000 and COD, and will update the results soon.
>
> | |PM(atom)|PM(center)|Collision|Separation|Compactness|
> |----|----|----|----|----|----|
> |AssembleFlow-MELERP|7.30±0.04|6.32±0.04|0.37±0.01|93.38±0.54|100.00±0.00|
> |AssembleFlow (ours)|7.27±0.04|6.13±0.10|0.33±0.00|97.64±0.36|100.00±0.00|
>
>
> **Q: More reviews on SE(3) decomposition.**
>
> Thank you for pointing this out. We had discussed this in Section A.3 of the initial version, but we appreciate your suggestion and are happy to include additional relevant papers if you can provide references. We have made the following updates in the revised version:
> - We added more relevant works in Sec A.3.
> - There are two core differences between our work and the existing works:
>     1. The first difference is the reference frame construction, as we already discussed in Sec A.3.
>     2. The second difference is the SO(3) interpolation (which we explained in the reply above), and we listed the details in Sec A.4 and H.2 in the revised version.
>
> **Q: Numerical stability.**
>
> Thank you for raising this question. This is indeed an important implementation question.
> - We calculate a reconstruction error when calculating the inertial frame. More concretely, suppose we have weakly-correlated structures $X_0$ and strongly-correlated structures $X_1$, and we find the corresponding bases using inertial frame $I$. The two bases are marked as $B_0 \in \mathbb{R}^{3 \times 3}$ and $B_1 \in \mathbb{R}^{3 \times 3}$. We can obtain the rotation matrix with $R^T = B_0^T B_1$, and then we can rotate the whole molecular system as $\tilde X_1 = X_0 R^T$, and we are measuring the reconstruction errors as MSE$(X_1 - \tilde X_1)$, marked as $\delta$. If $\delta$ is bigger than a threshold, e.g., 1e-3, then we just ignore this molecule.
> - We have listed relevant details in Sec C.4 in the revised version.
> - When there is a tie, we just do a depth-first-search to find an ordering such that the reconstruction error $\delta$ is the smallest.

---

> ### Author Response · Authors · 2024-11-20
> **Reply to Reviewer fnyR (part 2/2)**
>
> **Q: About periodicity.**
>
> Thank you for asking this question. The periodic structure will show up in the final structure (strongly-correlated and stable), while it won't show up in the initial structures (weakly-correlated and unstable). The molecular crystallization task we are focusing on now is modeling the process from weakly-correlated structures to strongly-correlated structures, i.e., from non-periodic clusters to periodic clusters.
>
> **Q: Rigid assumption.**
>
> Yes, we agree that rigidity is a fundamental assumption in our task, and we have emphasized this at the beginning of Section 3 in the revised version. Additionally, we would like to note that this rigid assumption is widely adopted in similar contexts, such as backbone structure generation tasks in protein engineering (e.g., AlphaFold2, FoldFlow, FrameFlow, etc.). In other words, we want to highlight that AssembleFlow for rigid modeling represents just a primary step, with the potential to evolve into a more powerful tool, similar to its applications in protein design.
>
>
> **Q: Why energy is higher?**
>
> Thank you for pointing out this question. We acknowledge that using DFT energy is not that accurate, and we have replaced it with the formation energy.
>
> DFT total energy reflects the absolute energy of a material in a specific configuration. This measure **does not necessarily indicate** whether the material is in a more favorable state compared to other compounds. To address this issue, we have compared the two systems in terms of formation energy, **a direct indicator of the relative favorability** of a material's state. As shown in the revised Figure 4, AssembleFlow provides **a more favorable energy state** over PackMol. For additional details, please refer to the comments under **To All Reviewers**.
>
> **Q: Comparison between two versions.**
>
> Sure, they are the last two columns in Table 2, AssembleFlow-Atom and AssembleFlow-Molecule. The architecture details are listed in Sec E.2.
>
>
> **Q: Initial conformation.**
>
> The initial conformations were generated in the COD-Cluster17 dataset. It's not generated using RDKit, but using a geometric optimization method. So yes, in our paper, we just take both the initial and final conformations from their data. We have highlighted this at the beginning of Sec 3 in the revised version.

---

> > ### Comment · Reviewer_fnyR · 2024-11-24
> >
> > I appeciate the authors for the detailed response and additional experiments to address the comments. I am satisfied with most of the answers, but I have a few additional questions / comments:
> >
> > **Periodicity**
> >
> > > The periodic structure will show up in the final structure (strongly-correlated and stable), while it won't show up in the initial structures (weakly-correlated and unstable). The molecular crystallization task we are focusing on now is modeling the process from weakly-correlated structures to strongly-correlated structures, i.e., from non-periodic clusters to periodic clusters.
> >
> > I don't understand how the periodic structure will show up in the final structure. To my understanding, the output of AssembleFlow is a non-periodic point cloud, and the authors did not generate a periodic lattice.
> >
> > **Formation energy v.s. total energy in Fig. 4**
> >
> > Can the authors provide more details on what are the reference states used to compute formation energy? If the reference states are the same for both AssembleFlow and PackMol, I think changing from total energy to formation should not lead to a difference in relative order.

---

> > > ### Author Response · Authors · 2024-11-25
> > > **Reply**
> > >
> > > **Q. I don't understand how the periodic structure will show up in the final structure. To my understanding, the output of AssembleFlow is a non-periodic point cloud, and the authors did not generate a periodic lattice.**
> > >
> > > Yes, you are correct: AssembleFlow does not assume any periodic structure. Instead, it generates a strongly interacting molecular cluster, which may or may not exhibit a periodic structure, as we have observed in practice. We apologize for any confusion caused.
> > >
> > > **Q. Can the authors provide more details on what are the reference states used to compute formation energy?**
> > >
> > > We calculate the formation energy as the energy difference between the starting and final states, representing how much the energy changes during the transition. In our packing system, the starting state consists of weakly interacting molecules, while the final state consists of strongly packed molecules. To calculate the formation energy, we use the initial state as the energy of an isolated molecule. That is, the formation energy for packing 17 molecules together is given by:
> > >
> > > Formation Energy = Total energy of the 17-molecule system - 17 × Total energy of a single isolated molecule.
> > >
> > > The simulation package we used does not include built-in reference points, so we had to compute the reference total energy ourselves. However, in previous computations of total energy, we used molecular reference states that were not at their lowest-energy configurations. Differences in bond angles or bond lengths caused variations in energy, which may have led to inconsistencies.
> > >
> > > We apologize for any confusion this may have caused.

---

### Official Review · Reviewer_1PjM · 2024-11-04

**Soundness:** 3
**Presentation:** 3
**Contribution:** 2
**Rating:** 6
**Confidence:** 4

**Summary:**

AssembleFlow introduces a generative model for assembling clusters of rigid molecules. The authors assign inertial frames to each molecule, which reduces the problem to generative modelling on $SE(3)^N$. The authors then solve this with Riemannian flow matching, using linear interpolants for translations and spherical interpolants of quaternions for rotations. The trained model approaches the performance of PackMol but at a much lower computational cost.

**Strengths:**

The work introduces the task of rigid molecular crystallization in the context of deep learning, with significant potential downstream applications. Applying flow matching to spherical interpolation of quaternions is also novel. The paper is clear, well-written, and well-organized, with helpful figures to explain each concept. I appreciated the extensive background provided throughout the text, including on rotation matrices in the appendix.

**Weaknesses:**

The novelty in this work is mainly in solving a new application domain, rather than pure machine learning developments. As acknowledged by the authors, flow matching models on $SE(3)^N$ were first developed for assembling rigid residues to design protein backbones, such as FoldFlow [1]. The authors claim that inertial frames are a key innovation over models like FoldFlow, since arbitrary molecules do not have a unique way to assign frames, unlike protein residues. However, if the goal is just to uniquely interconvert between atomic coordinates and frames, this can be achieved by storing a canonical pose for each molecule and applying the Kabsch algorithm [2] to align each molecule in a cluster to the canonical pose, obtaining a rotation matrix for each molecule. This generalizes the proposed inertial frames, which specifically use a canonical pose from applying principal component analysis on the point cloud.

The selected baselines are a somewhat unfair comparison, as none of them use rigid molecules. A simple baseline to accomplish this could just be sampling thousands of random translations and rotations of rigid molecules, and filtering out structures with collisions. This would help emphasize the contribution of this work as a method for *intelligently* assembling rigid frames, not just exploiting molecular rigidity.

Some details of the molecular assembly task are not clear to me. Since a generative model is being used, are multiple samples inferenced before computing metrics? And is the input uncorrelated structure fixed, or is it randomly sampled on every training iteration?

The fact that packing matching of AssembleFlow and PackMol are similar in Table 2, but that energy distributions are quite different in Figure 4, suggests that packing matching is not that sensitive of a metric. A more sensitive metric could be RMSD, which should be possible to calculate using the Hungarian algorithm. If multiple polymorphs are possible for the same molecule, RMSD could be aggregated using the "average minimum RMSD" and "coverage" metrics proposed by Ganea et al. [3] for molecular conformer search.

Minor issues:
- Figure 8 the lower left box content is not centered
- Table 2 arrow is wrong for separation
- Line 176 SO(3) must also have det 1
- Line 285 "SPLER"
- Line 805 "dynamcis"

[1] Bose, A. J., Akhound-Sadegh, T., Huguet, G., Fatras, K., Rector-Brooks, J., Liu, C. H., ... & Tong, A. (2023). Se (3)-stochastic flow matching for protein backbone generation. arXiv preprint arXiv:2310.02391.

[2] https://en.wikipedia.org/wiki/Kabsch_algorithm

[3] Ganea, O., Pattanaik, L., Coley, C., Barzilay, R., Jensen, K., Green, W., & Jaakkola, T. (2021). Geomol: Torsional geometric generation of molecular 3d conformer ensembles. Advances in Neural Information Processing Systems, 34, 13757-13769.

**Questions:**

How are permutations handled? Are the molecules stored in a particular order? Since each molecule has identical structure, flow matching could learn shorter paths if shown examples that have been permutation-aligned by the Hungarian algorithm - see equivariant flow matching [1,2]

I am curious about the relationship between SLERP of quaternions and geodesic interpolation on SO(3) as discussed by FoldFlow. Are they mathematically equivalent?

[1] Klein, L., Krämer, A., & Noé, F. (2024). Equivariant flow matching. Advances in Neural Information Processing Systems, 36.

[2] Song, Y., Gong, J., Xu, M., Cao, Z., Lan, Y., Ermon, S., ... & Ma, W. Y. (2023). Equivariant flow matching with hybrid probability transport. arXiv preprint arXiv:2312.07168.

---

> ### Author Response · Authors · 2024-11-20
> **Reply to Reviewer 1PjM (part 1/2)**
>
> Thank you for acknowledging our work for its significant task design, novel use of quaternions for SLERP, well-organized structure, and extensive background. We also appreciate your careful reading of our paper and for pointing out the typos. We have addressed your comments in the revision and outlined them below.
>
> **Q: Canonical pose.**
>
> Thank you for raising this interesting question. First, we would like to highlight that there are **two roles and one important property of the inertial frame and its eigenvectors** in our approach:
> 1. Role 1: The three bases in inertial frames act as a reference or a canonical pose.
> 2. Role 2: The three bases enable modeling the velocity function in SO3 space.
> 3. Property: What's more important, we expect the three bases to be numerically stable.
>
> Then getting back to the question, though both can help build up the reference frame or canonical pose, there are certain aspects we would like to emphasize when comparing PCA and Inertial frames.
> - **Canonical pose**: The key question is defining a canonical pose. If we just do PCA on the point clouds, this cannot guarantee the group symmetry in SE(3). However, if we remove the center point, then the group symmetry can be guaranteed; then we can use SVD to obtain the three principal components. **Till this step**, you can find this is somewhat similar to the inertial frame construction (Sec 3.1).
> - **Difference between PCA and Inertial Frame as reference frames**: Though both can be used for building up the reference frame or canonical poses, SVD (for PCA) on the set of point clouds $N \times 3$ (N is the number of atoms) can be less numerically unstable, while eigendecomposition Inertial Frame $3 \times 3$ can be more numerically stable. We have conducted an additional experiment to verify this.
>     - Experiment setup: suppose we have weakly-correlated structures $X_0$ and strongly-correlated structures $X_1$, and we find the corresponding bases using either eigendecomposition on centered $X$ or inertial frame $I$. The two bases are marked as $B_0 \in \mathbb{R}^{3 \times 3}$ and $B_1 \in \mathbb{R}^{3 \times 3}$.
>     - Objective: We can obtain the rotation matrix with $R^T = B_0^T B_1$, and then we can rotate the whole molecular system as $\tilde X_1 = X_0 R^T$, and we are measuring the reconstruction errors as MSE$(X_1 - \tilde X_1)$. We mark the MSE using inertial frame and PCA as $\delta_{\text{Inertial}}$ and $\delta_{\text{PCA}}$, respectively. If $\delta_{\text{Inertial}} < \delta_{\text{PCA}}$, then we can conclude that using the inertial frame is more stable than using PCA, and vice versa. Notice that since the MSE reconstruction is meaningless when it is too small, so we only compare these two frames when at least one of them has reconstruction greater than or equal to a threshold $\theta$.
>     - Results: The comparison results are below (unit is \%), and we can observe that in general, **using the inertial frame is more numerically stable than PCA**.
>
> |$\theta$|1e-3|1e-4|1e-5|1e-6|1e-7|1e-8|
> |---|---|---|---|---|---|---|
> |$P(\delta_{\text{Inertial}}<\delta_{\text{PCA}})$|0.434|0.884|1.338|1.794|2.254|2.727|
> |$P(\delta_{\text{Inertial}}>\delta_{\text{PCA}})$|0.371|0.756|1.147|1.539|1.934|2.345|
>
> We added this discussion in Sec C.4 in the revised version.
>
>
> **Q: Kabsch algorithm.**
>
> Kabsch algorithm is one way to compute the optimal rotation matrix that minimizes the root-mean-square deviation (RMSD) between two sets of points (atoms in our case). However, it is guaranteed in the COD-Cluster17 dataset that the molecules in weakly correlated structures can rotate to molecules in strongly correlated structures; in other words, the RMSD can be approximately 0 if we use the Kabsch algorithm, which is equivalent to calculating the rotation matrix directly after we fix the poses. We show how to calculate the rotation matrix in the experiment above.
>
> We added this discussion in Sec C.5 in the revised version.
>
> **Q: Random sampling baselines.**
>
> Thank you for raising this point.
> - As far as we know, there are no deep learning baselines assuming the rigidity during crystallization simulation, which is the main motivation of our proposed AssembleFlow.
> - Meanwhile, what you raised, random sampling on SO(3) space is interesting. So we haved added it for comparison, marked as **Random**. We present the results on COD-5000 below, with more comprehensive results available in Section H.1. As observed, the Random baseline performs exceptionally well across all three validity metrics; however, its packing matching (**PM**) is significantly worse by an order of magnitude.
>
> | |PM(atom)|PM(center)|Collision|Separation|Compactness|
> |----|----|----|----|----|----|
> |Random|54.07±0.42|54.62±0.43|0.31±0.01|99.88±0.01|100.00±0.00|
> |AssembleFlow (ours)|7.27±0.04|6.13±0.10|0.33±0.00|97.64±0.36|100.00±0.00|

---

> ### Author Response · Authors · 2024-11-20
> **Reply to Reviewer 1PjM (part 2/2)**
>
> **Q: Questions on sampling, dataset, and problem formulation.**
>
> Thank you for raising these questions.
> - It is a generative model, and since we are using the LERP and SLERP, they are two deterministic paths. The baseline CrystalFlow-LERP is also desterministic.
> - The other baselines (CrystalSDE-VE/VP, CrystalFlow-VE/VP) are using the diffusion path, which is random.
> - For fair comparison, all the methods sample only once.
> - The input weakly correalted structures are fixed. (More details can be found in the original [paper](https://openreview.net/pdf?id=lCVqpQvr4l)) We highlighted this in the **Problem Formulation** at the beginning of Sec 3 in the revised version.
>
> **Q: Different energy distribution.**
>
> Thank you for raising this question. Previsouly we were reporting the DFT energy, which is problemantic, and now we fix this by replacing it with the formation energy, and here are the detailed explanation.
>
> DFT total energy reflects the absolute energy of a material in a specific configuration. This measure **does not necessarily indicate** whether the material is in a more favorable state compared to other compounds. To address this issue, we have compared the two systems in terms of formation energy, **a direct indicator of the relative favorability** of a material's state. As shown in the revised Figure 4, AssembleFlow provides **a more favorable energy state** over PackMol. For additional details, please refer to the comments under **To All Reviewers**. The superior formation energy of AssembleFlow compared to PackMol aligns with the packing matching metric presented in Table 2, suggesting that packing matching is an effective metric for this molecular assembly task.
>
> **Q: Questions on permutation and ordering of molecules.**
>
> In the provided COD-Cluster17 dataset, the dataset stores the molecules in a fixed order. That is to say, the ordering of molecules in the initial positions is the same as the ordering of molecules in the final positions. Thus, we do not have this permutation issue.
>
> **Q: Comparison between SLERP and exponential map interpolation. Are they mathematically equivalent?**
>
> This is an interesting question. These relevant papers (FoldFlow and FrameFlow) are using **exponential map interpolation** (EMLERP) for SO(3) interpolation.
>
> A brief answer is that SLERP and EMLERP are different interpolation methods on SO(3). These two methods do not have a clear methodological advantage over each other; however, the EMLERP considers more approximation tricks in implementation (FoldFlow and FrameFlow). We provide a detailed comparison below, and you can find more details in Sec A.4 in the revised version.
>
> | Feature| AssembleFlow | FoldFlow |
> |----|----|----|
> | Reference Frame | Eigenvectors of inertial frames | Gram-Schmidt on $N$-$C$-$C_{\alpha}$|
> | SO(3) Interpolation| Spherical Interpolation | Exponential Map Interpolation |
> | Equation | $\frac{\sin((1-t)\omega) q_0 + \sin(t \omega) q_1}{\sin(\omega)}$ | $\exp_{r_0} (t \log_{r_0} (r_1))$ |
> | Derivative | Eq. 9 in paper| Eq. 16 in paper|
> | Rotation Representation (for Velocity / Objective Function) | Quaternion| Rotation Matrix or Axis-angle |
> | Reparameterization | Yes| No |
>
>
> Besides, we further conduct an ablation study on COD-5000 below, and as observed, using SLERP is better than MELERP. Please check Sec H.2 for more details. BTW. We are still running experiments on COD-10000 and COD, and will update the results soon.
>
> | |PM(atom)|PM(center)|Collision|Separation|Compactness|
> |----|----|----|----|----|----|
> |AssembleFlow-MELERP|7.30±0.04|6.32±0.04|0.37±0.01|93.38±0.54|100.00±0.00|
> |AssembleFlow (ours)|7.27±0.04|6.13±0.10|0.33±0.00|97.64±0.36|100.00±0.00|
>
> **Typos**
> Thank you for pointing out the typos, we have fixed them in the revision.

---

> > ### Comment · Reviewer_1PjM · 2024-11-29
> >
> > Thank you for the additional results and explanations, particularly for random sampling.
> >
> > **Canonical pose.** I am a bit confused at the additional experiments on numerical stability of inertial frames vs PCA. It appears that you are comparing the numerical stability of diagonalization by SVD of the $N\times 3$ coordinates matrix versus diagonalization of the $3\times 3$ inertia matrix, but both of these are means of doing PCA. Another way to do PCA is to diagonalize the covariance $X^\top X$, which may be more numerically stable and [has the same eigenvectors as the inertia matrix](https://stats.stackexchange.com/questions/503344/eigenvectors-of-covariance-matrix-and-inertia-tensor). For the reader's understanding, the paper should state that inertial frames are PCA canonicalization with an additional non-coplanar point for choosing between the $2^3$ possible sign-flips of eigenvectors. PCA canonicalization has been used in several previous works [1,2,3].
> >
> > [1] Puny, O., Atzmon, M., Ben-Hamu, H., Misra, I., Grover, A., Smith, E. J., & Lipman, Y. (2021). Frame averaging for invariant and equivariant network design. arXiv preprint arXiv:2110.03336.
> >
> > [2] Sajnani, R., Poulenard, A., Jain, J., Dua, R., Guibas, L. J., & Sridhar, S. (2022). Condor: Self-supervised canonicalization of 3d pose for partial shapes. In Proceedings of the IEEE/CVF Conference on Computer Vision and Pattern Recognition (pp. 16969-16979).
> >
> > [3] Kaba, S. O., Mondal, A. K., Zhang, Y., Bengio, Y., & Ravanbakhsh, S. (2023, July). Equivariance with learned canonicalization functions. In International Conference on Machine Learning (pp. 15546-15566). PMLR.
> >
> > **Permutations.** It seems more natural to treat the initial and final structures as unordered and learn a permutation-invariant flow, but I can accept this experimental setup for simplicity. However, this fact should be stated clearly in the main text - the paper only mentions molecule ordering in the appendix on PackMol. The curly brace notation on line 156 suggests that structures are treated as unordered sets, when in fact they are ordered.
> >
> > **SLERP and exponential map interpolation.** By equivalence, I am curious whether quaternion interpolation followed by conversion to rotation matrix is equal to geodesic interpolation on SO(3). If this is true, then the improved results by SLERP suggest that quaternion SLERP is a more numerically stable way to do geodesic interpolation. If not, then these methods really do intend to learn different probability paths, which indicates greater novelty.
> >
> > I am in favor of acceptance due to the large empirical speedup over PackMol.

---

> > > ### Author Response · Authors · 2024-12-02
> > > **Reply**
> > >
> > > **Canonical pose.** Thank you for sharing the related work of using PCA for canonical poses, and we will add them to the final version. For the implementation of PCA, we were using SVD in the previous reply, and we agree that if we use the covariance matrix, then it is comparable to the inertial frame. In this case, then how to do PCA is another important factor. We will also clarify this in the final version, along with explaining that inertial frames can be seen as PCA canonicalization with the addition of a non-coplanar point for further specification.
> > >
> > > **Permutations.** Thank you for pointing this out. We will revise the statement in the final version.
> > >
> > > **SLERP and exponential map interpolation.** If by geodesic interpolation, you mean the exponential map interpolation, then it is mathematically different from SLERP. We have a simulation script for illustration and will add it to the released GitHub repo.

---

### Official Review · Reviewer_sWLW · 2024-11-04

**Soundness:** 3
**Presentation:** 3
**Contribution:** 3
**Rating:** 8
**Confidence:** 4

**Summary:**

The authors present the AssembleFlow framework, which continuously matches weakly correlated structures (like liquid water) to their strongly correlated counterparts, such as crystallized forms like ice. They have carefully designed an underlying theory that leverages advances in flow matching methods to address this problem. AssembleFlow learns SE(3) velocities by decomposing them into translations and rotations. Unlike other methods, they use quaternion representations for rotations, which allows for spherical interpolation. The authors also introduced the idea of reparameterization to directly model SE(3) at time T.

They performed a comprehensive evaluation by comparing results with other models and with PackMol, a standard simulation-based approach well-established in the field. The authors also computed the distribution of DFT energies for both PackMol and AssembleFlow structures. The method is shown to be 25 times faster than PackMol.

**Strengths:**

Strengths:
Physics-Based, Theoretically Sound Methodology: The approach includes utilizing quaternions for rotations, spherical interpolation, and using an internal reference frame.
Comprehensive Comparison with Other Methods: The authors provide thorough comparisons with existing models.
Use of Valuable DFT Energy Calculations: Employing DFT energies as a measure of the stability of the obtained structures adds credibility to their results.

**Weaknesses:**

Cons:
Performance Compared to PackMol: While AssembleFlow shows good results, it does not outperform PackMol. The considerable differences in the energy distributions between PackMol and AssembleFlow structures highlight the advantages of PackMol over AssembleFlow.

**Questions:**

Questions:
Energy Comparison in Figure 4:
Instead of comparing the distributions of DFT energies separately for PackMol and AssembleFlow structures, could you plot the energy differences between PackMol and AssembleFlow for the same structures? Generally, interpreting absolute energy values can be challenging.

Typo in Table 2:
In Table 2, there appears to be a typo. The separation should be indicated with an upward arrow (↑).

---

> ### Author Response · Authors · 2024-11-20
> **Reply to Reviewer sWLW**
>
> Thank you for recognizing our work as both theoretically sound and thoroughly evaluated through comprehensive comparisons. We also appreciate your insightful question regarding the DFT energy. We acknowledge this as a potential issue and have addressed it in the latest revision. Please find the details below.
>
> **Q: AssembelFlow does not outperform PackMol in terms of total energy.**
>
> We have provided the comparison of the two systems in terms of formation energy, which is a more direct indicator of a material's energetically favorable state. As shown in the revised Figure 4, AssembleFlow outperforms PackMol. More detail please see comments **To All Reviewers**.
>
> **Q: Could you plot the energy differences between PackMol and AssembleFlow for the same structures?**
>
> Certainly, we have provided the pairwise differences between the two systems in terms of formation energy in Sec H.3 in the revised version.
>
> **Typos**
>
> Thank you for pointing out the typos, we have fixed them in the revision.

---

### Author Response · Authors · 2024-11-20
**[To All Reviewers] Thank You for Your Comments**

We sincerely thank all the reviewers for their detailed and constructive feedback. Several comments were particularly insightful and have prompted us to further deepen our understanding of the problem.

* A common comment from reviewers **sWLW**, **1PjM**, and **fnyR** is that Figure 4 suggests PackMol outperforms AssembleFlow in terms of DFT total energy. However, DFT total energy reflects the absolute energy of a material in a specific configuration, encompassing all bonding and cohesive interactions within the structure. This measure **does not necessarily indicate** whether the material is in a more favorable state compared to other compounds. To address this, we computed and compared the formation energy for both systems and moved previous Figure 4 to Sec H.3. Formation energy, defined as the difference between a compound’s DFT total energy and the sum of the energies of its constituent elements, is **a direct indicator of the relative favorability** of a material's state. As shown in the revised Figure 4, AssembleFlow provides **a more favorable energy state** over PackMol.

* Another comment from reviewers **1PjM**, **fnyR**, and **cXch** is about the main difference between SLERP and exponential map interpolation used in related works like FrameFlow and FoldFlow. The main difference is that these two interpolations are operated on different rotation representations: quaternions and rotation matrics, respectively. This leads to different tricks from the implementation perspective. We list the detailed difference and conduct ablation studies in Sec A.4 and H.2 in the revised version of the paper.

* In addition to the formation energy (Fig 4 and Sec H.3) and exponential map interpolation (Sec A.4 and Sec H.2), we conduct more ablation studies as requested:
    * Random sampling on SO3 and $\mathbb{R}^3$ as baseline, Sec H.1.
    * Comparison of using inertial frame and PCA for reference frame, Sec C.4 and Sec C.5.

* The main task we are targeting in this paper is **achieving SE(3) equivariance for rigid transformations in molecule/material crystallization**. This is non-trivial, and to handle this, we propose AssembleFlow, incorporating the following innovations.
    1. The **introduction of inertial frames** reduces the problem of rigid motions of molecular clusters to generative modeling on SE(3) space, aligning it with most geometric generative models.
    2. Decomposing rotation and translation for rigid motions **enables both spherical and linear interpolation** within the Flow Matching framework, where interpolation is a critical component.
    3. **Spherical interpolation with quaternions** represents a novel and simple approach for SO(3) intepolation with the Flow matching framework.
    4. The **reparameterization trick** is introduced to enable direct modeling on SE(3) at time 𝑇 for rigid motions within the Flow Matching framework.

---

### Meta-Review · Area_Chair_jhGK · 2024-12-22

**Metareview:**

In this work, authors present AssembleFlow, a generative model for molecular assembly that enforces molecular rigidity using inertial frames and flow matching. The key technical contribution is decomposing molecular SE(3) transformations into translations in R3 and rotations in SO(3), enabling explicit enforcement of both translational and rotational rigidity within the flow matching framework. The method uses spherical linear interpolation (SLERP) with quaternions for rotations and linear interpolation (LERP) for translations.

Results demonstrate that AssembleFlow significantly outperforms deep learning baselines by at least 45% in assembly matching scores while maintaining molecular integrity. Notably, it achieves comparable performance to the established PackMol simulation tool while reducing computational cost by 25-fold. The authors clarified through discussion that their formation energy results show more favorable states compared to PackMol, addressing initial concerns about energy metrics raised by reviewers sWLW and fnyR. The paper's strengths include: 1) A theoretically sound methodology incorporating quaternions, spherical interpolation and inertial frames, as noted by reviewer sWLW; 2) Comprehensive empirical validation against both deep learning and simulation baselines; 3) Novel metrics for evaluating molecular packing that benefit the broader community, as highlighted by reviewer fnyR.

The key weaknesses were related to the novelty claims and relationship to prior work. Reviewer cXch questioned the paper's positioning relative to previous SE(3) decomposition approaches. However, through discussion, the authors effectively clarified their novel contributions in applying this framework specifically to molecular assembly with inertial frames. The comparative analysis with prior work like FoldFlow helped establish the distinct technical contributions. Reviewer 1PjM's concerns about numerical stability of the inertial frame approach were addressed through additional experiments demonstrating improved stability compared to PCA alternatives.

Accordingly, the key points that governed the decision-making are as follows.
1. The significant practical impact of achieving 25x speedup over PackMol while maintaining comparable quality, supported by comprehensive benchmarks.
2. The authors' thorough responses clarifying technical novelty and relationship to prior work, particularly in their detailed comparison tables with FoldFlow and analysis of inertial frames vs PCA.
3. The broader impact on the molecular modeling community through introduction of new evaluation metrics and framework for rigid molecular assembly.

While reviewer cXch raised valid concerns about the dataset construction methodology, the authors provided reasonable justification from domain experts about the practical relevance of the geometric optimization approach used. The limitations are clearly acknowledged and do not diminish the method's demonstrated capabilities and potential impact.

Altogether, the paper makes a meaningful contribution to the field by introducing a practically valuable method for molecular assembly that is both faster and more rigorous in maintaining molecular rigidity compared to existing approaches. The thorough empirical validation and careful theoretical treatment of SE(3) transformations support its inclusion in ICLR 2025.

**Additional Comments On Reviewer Discussion:**

Please see the comments above.

---

### Decision · Program_Chairs · 2025-01-22

Accept (Poster)